# Towards Reliable Spatiotemporal Epidemic Forecasting via Steering Diffusion Inference

## Abstract

Reliable epidemic prediction is vital for public health response and resource allocation, especially in rapidly evolving outbreaks. Despite the recent attempts to integrate the epidemic mechanistic model into data-driven forecasting models, existing approaches still lack interpretability and robustness. To bridge this gap, we propose `EpiDiff`, an epidemiology-aware diffusion framework that incorporates mechanistic estimations and their posterior uncertainties into the forecasting process. `EpiDiff` features a flexible and high-capacity diffusion backbone specifically designed for spatio-temporal epidemic data, enabling accurate and robust sequence prediction. By quantifying the uncertainty of mechanistic forecasts and using it to steer the diffusion model at inference, `EpiDiff` dynamically adjust the data-driven prediction with the guidance from epidemic model. Extensive experiments on real-world epidemic datasets demonstrate that `EpiDiff` consistently outperforms state-of-the-art baselines in both accuracy and robustness, while offering improved explainability for epidemic forecasting. Our code and datasets are available at https://anonymous.4open.science/r/epidiff-4782.

## 1 Introduction

In recent years, outbreaks of infectious diseases such as COVID-19 have underscored the urgent need for accurate modeling and forecasting of epidemic dynamics (Flaxman et al., 2020; Chinazzi et al., 2020; Block et al., 2020; Pei et al., 2020). Reliable forecasts are critical for informing policymakers about effective interventions and for guiding the allocation of limited healthcare resources (Reich et al., 2019). Existing epidemic forecasting approaches can be generally divided into two categories: model-based (mechanistic) and data-driven methods. Commonly used mechanistic models include patch-based and network-based compartmental frameworks (Hethcote, 2000; Pastor-Satorras et al., 2015), which rely on explicit assumptions about transmission dynamics, such as the Susceptible–Infectious–Recovered (SIR) framework (Kermack & McKendrick, 1927; Anderson, 1991), and typically use systems of differential equations with calibrated parameters to simulate future epidemic trajectories (Chowell et al., 2020; Jewell et al., 2020). The data-driven methods range from traditional statistical approaches such as autoregressive (AR) and autoregressive moving average (ARMA) models (Box et al., 2015; Brockwell & Davis, 2002), to state-space formulations such as Kalman filters Kalman (1960), and deep learning models (Chimmula & Zhang, 2020; Kapoor et al., 2020; Wang et al., 2020). These methods learn patterns from historical epidemiological data through statistical modeling or complex spatiotemporal architectures to generate forecasts.

In the context of emerging epidemics such as COVID-19, forecasting faces several critical challenges (Ioannidis et al., 2022; Jewell et al., 2020): (1) surveillance data is often sparse due to the absence of long-term historical records, which could restrict capacity of data-driven models to adequately train predictive models (Delussu et al., 2023). In addition, their dependence on historical patterns renders them fragile to distributional shifts induced by sudden outbreaks or behavioral changes, leading to out-of-distribution scenarios that undermine predictive accuracy (Kapoor et al., 2020); and (2) the disease spread process reflects the rapidly co-evolving interplay of individual behavior, government interventions, and disease transmission rate (Chang et al., 2021; Haug et al., 2020), which cannot be adequately captured by a single epidemic model(Stocks et al., 2020; Castro et al., 2021).

One promising solution to address these challenges is to develop hybrid epidemic forecasting models that merge neural networks' predictive power with mechanistic models' fundamental principles (Liu et al., 2024). Recent approaches have begun integrating epidemiological domain knowledge with neural architectures (Song et al., 2020b), such as using compartmental model states as causal features (Wang et al., 2022) or employing SIR-inspired neural ODE formulations (Wan et al., 2025). However, existing hybrid models primarily focus on improving prediction accuracy and face two critical limitations: (1) they lack explainable mechanisms to quantify the respective contributions of mechanistic and data-driven components in the final prediction, making it difficult to explain the source and reliability of forecasts; and (2) they lack sufficient flexibility to handle the uncertainty and non-stationarity inherent in epidemic systems, where surveillance data are noisy and model parameters evolve over time (Holmdahl & Buckee, 2020; Stocks et al., 2020). The limitations are illustrated in Fig. 1.

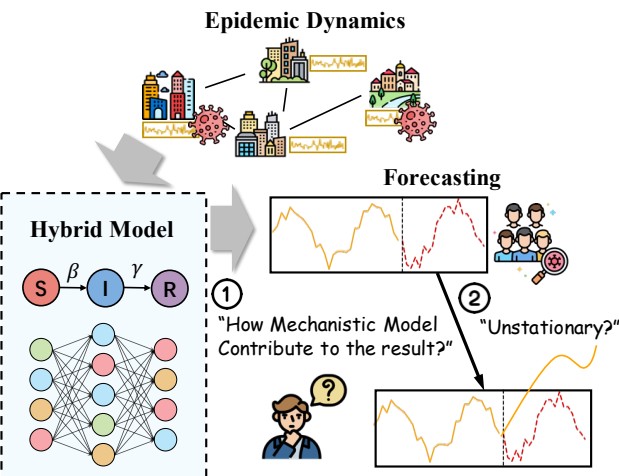

Figure 1: An illustration of Challenges of Hybrid Epidemic Forecasting Models.

In this paper, we address the challenges of explainability and robustness in hybrid epidemic forecasting models by developing a novel unified epidemiology-aware spatiotemporal forecasting framework based on diffusion models. We choose diffusion models as the backbone data-driven approach due to their demonstrated prominence in spatiotemporal forecasting across diverse domains, including transportation (Wen et al., 2023; Zheng et al., 2023) and crime risk prediction (Wang et al., 2024). Our approach quantifies the uncertainty in mechanistic model estimates, enabling the diffusion model to be heterogeneously guided by mechanistic estimations across different spatiotemporal regions during inference. This design produces robust and explainable epidemic forecasts that leverage both mechanistic understanding and data-driven flexibility. Our key contributions are as follows:

- **Unified Hybrid Framework**: We propose `EpiDiff`, a unified epidemiology-aware spatiotemporal forecasting framework that integrates mechanistic models with diffusion-based deep learning, bridging the gap between explainability and predictive accuracy in epidemic forecasting.

- **Uncertainty-Aware Mechanistic Guidance**: We introduce a principled mechanism to quantify the posterior uncertainty of mechanistic estimations, which dynamically modulates the influence of mechanistic estimations in the diffusion inference process, leading to robust and adaptive predictions across heterogeneous spatiotemporal regions.

- **Comprehensive Empirical Validation**: Extensive experiments on real-world COVID-19 and influenza datasets show that `EpiDiff` consistently outperforms state-of-the-art baselines in both forecasting accuracy and robustness, while providing explainable, uncertainty-aware predictions.

## 2 PRELIMINARY

### 2.1 PROBLEM FORMULATION

Let $\mathcal{G} = (\mathcal{V}, \mathcal{E}, \mathbf{A}, \mathbf{X})$ denote an attributed graph, where $\mathcal{V}$ is the set of nodes with $|\mathcal{V}| = N$, and $\mathcal{E} \subseteq \mathcal{V} \times \mathcal{V}$ is the edge set describing spatial relations. The adjacency matrix is $\mathbf{A} \in \mathbb{R}^{N \times N}$, and $\mathbf{X} \in \mathbb{R}^{N \times C}$ is the node feature matrix. Each row $\mathbf{X}_i$ contains multiple attributes of region $v_i$, such as epidemic indicators and auxiliary covariates.

**Definition 1** (Spatio-Temporal Graph). *A spatio-temporal graph $\{\mathcal{G}^t\}_{t=1}^T$ is a sequence of graphs indexed by time $t$, where each $\mathcal{G}^t = (\mathcal{V}, \mathcal{E}, \mathbf{A}, \mathbf{X}^t)$ and $\mathbf{X}^t \in \mathbb{R}^{N \times C}$ denotes node features at time $t$. The adjacency $\mathbf{A}$ encodes spatial dependencies, while temporal features capture disease dynamics and related covariates across $T$ steps.*

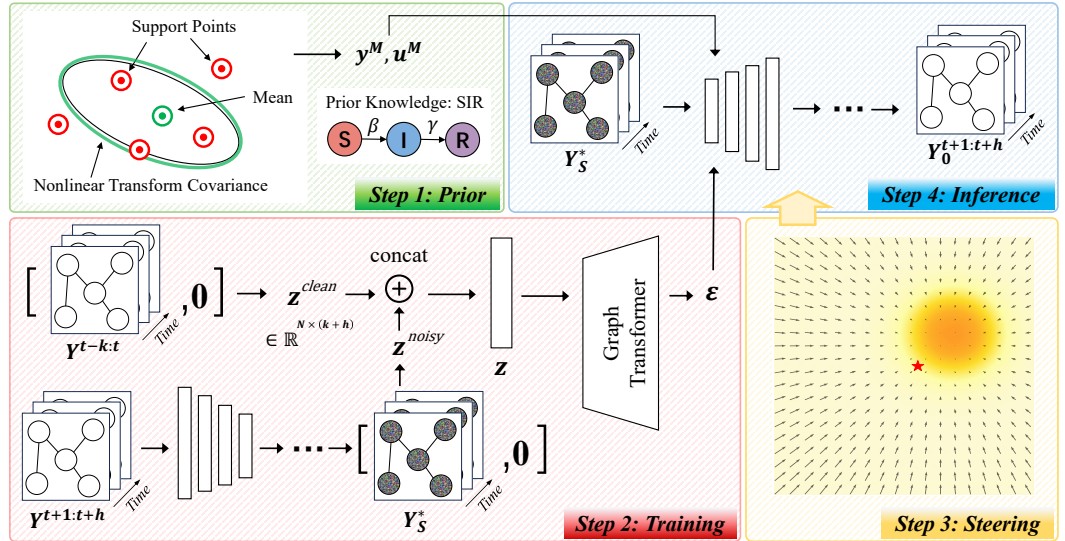

Figure 2: Overview of the `EpiDiff` pipeline. **Step 1: Prior**. We first fit a mechanistic SIR model to the historical epidemic trajectory and generate sigma-point-based posterior estimations of the future, including both the mean trajectory $\mathbf{y}^M$ and its uncertainty $\mathbf{u}^M$. **Step 2: Training**. We train a spatio-temporal diffusion backbone using observed epidemic data, which learns to generate future trajectories conditioned on past windows and graph structure. **Step 3: Steering**. During inference, we inject mechanistic prior information: $\mathbf{y}^M$ acts as a soft target and $\mathbf{u}^M$ modulates the guidance strength, enabling adaptive, uncertainty-aware generation. **Step 4: Inference**. The diffusion model generates future trajectories from noise, conditioned on history, graph structure, and dynamically steered by the mechanistic estimation.

We focus on predicting a subset of node features related to epidemic outcomes. Let $\mathbf{Y}^t \in \mathbb{R}^{N \times C'}$ ($C' < C$, in our paper we set $C' = 1$) denote the epidemic features (e.g., cases) extracted from $\mathbf{X}^t$.

**Problem 1** (Spatio-Temporal Epidemic Forecasting). *Given a sliding window of $\kappa$ historical steps $\mathcal{G}^{t-\kappa:t}$, the goal is to predict future epidemic features $\mathbf{Y}^{t+1:t+h}$, where $h$ is the prediction horizon.*

Unlike standard spatio-temporal forecasting, epidemic features follow mechanistic compartmental dynamics with heterogeneous parameters across nodes and time. To capture this, we formulate the following task.

**Problem 2** (Epidemic Forecasting with Dynamic Mechanistic Guidance). *Given $\mathcal{G}^{t-\kappa:t}$, assume each epidemic feature $y_i^t \in \mathbf{Y}^t$ of node $v_i$ follows an epidemic model $\Phi$ with parameters $\Theta_i^t$ that vary by node and time. The task is to predict $\mathbf{Y}^{t+1:t+h}$ while ensuring consistency with the mechanistic dynamics governed by $\Phi$.*

## 2.2 SIR DYNAMICS

As a representative mechanistic model, we adopt the susceptible–infected–recovered (SIR) system (Kermack & McKendrick, 1927; Keeling & Rohani, 2008). In this formulation, the total population $N$ is partitioned into three time-dependent compartments: $S$ denotes the number of susceptible individuals, $I$ the infected, and $R$ the recovered. The evolution of these states is described by the system as follows:

$$\frac{dS}{dt} = -\beta SI, \quad \frac{dI}{dt} = \beta SI - \gamma I, \quad \frac{dR}{dt} = \gamma I, \tag{1}$$

where $\beta$ is the transmission rate and $\gamma$ the recovery rate. This formulation links interpretable parameters $(\beta, \gamma)$ to observable new cases. We use the SIR system as a baseline mechanistic prior for guiding data-driven inference in our framework.

## 3 METHODOLOGY

In our framework, we focus on predicting new cases as representative epidemic feature with the guidance of SIR dynamics. We first introduce a metric that quantifies mechanistic uncertainty based on the mechanistic estimation error, then present a parameterized and efficient method for estimating mechanistic model parameters . Next, we detail the architecture and training objectives of our graph diffusion backbone. Finally, we present how mechanistic estimation and uncertainty characterization are used to provide heterogeneous guidance during diffusion model inference. The overall framework is illustrated in Fig. 2.

### 3.1 UNCERTAINTY QUANTIFICATION OF MECHANISTIC ESTIMATIONS

In an epidemic system with a given infection mechanism (e.g., SIR, SIS), epidemic dynamics are governed by deterministic differential equations (Kermack & McKendrick, 1927). Over short horizons, the model parameters (e.g., infection and recovery rates) can be regarded as stable. However, because real-world observations are noisy and incomplete, estimating these parameters is inherently uncertain. This estimation error can be summarized by a covariance matrix over the parameters, which quantifies the difficulty of accurate inference. When such parameter uncertainty is propagated through the mechanistic model, it induces variability in the predicted epidemic trajectories. We refer to this induced predictive variability as *mechanistic uncertainty*.

**Definition 2** (Mechanistic Uncertainty). *Let $\theta$ denote the parameter vector of a mechanistic epidemic model $\Phi$, with point estimate $\hat{\theta}$ and covariance $\Sigma$ obtained from parameter inference. For node $v_i$ and time $t$, the mechanistic uncertainty is defined as the predictive variance*

$$u_{i,t}^M = \text{Var}_{\theta \sim \mathcal{N}(\hat{\theta}, \Sigma)}[\Phi(\theta)_{i,t}], \tag{2}$$

*where $\Phi(\theta)_{i,t}$ denotes the model's forecast for node $v_i$ at time $t$. Thus, mechanistic uncertainty measures how parameter estimation error translates into variability of epidemic forecasts.*

The most common approaches for quantifying uncertainty in parameter estimation include Approximate Bayesian Computation (ABC) (Pritchard et al., 1999; Marjoram et al., 2003) and Markov chain Monte Carlo (MCMC) techniques such as Hamiltonian Monte Carlo (HMC) (Neal et al., 2011; Brooks et al., 2011). These methods approximate the posterior distribution but are computationally intensive, particularly for high-dimensional or time-sensitive forecasting tasks.

To efficiently approximate the posterior distribution of the parameters for each node, we adopt a Gaussian parameterization of the posterior. Specifically, we work in the transformed parameter space $\theta_i = (\log R_{0,i}, \log(1/\gamma_i))$, where $R_{0,i} = \beta_i/\gamma_i$ is the the basic reproduction number. This log-transformation ensures positivity of the original epidemiological parameters and provides a numerically stable representation. Moreover, $R_{0,i}$ and $1/\gamma_i$ (the mean infectious period) are epidemiologically interpretable parameters that have been extensively studied in public health and clinical research. This choice of parameterization allows us to directly incorporate domain knowledge into the prior distribution $p(\theta_i)$ and ensures the plausibility of the parameter estimates.

We define the negative log-posterior of node $i$ as

$$\mathcal{J}_i(\theta_i) = -\log p(y_i^{t-\kappa:t-1} \mid \theta_i) - \log p(\theta_i), \tag{3}$$

where $y_i^{t-\kappa:t-1}$ denotes the epidemic features of node $i$ observed in the time window. The posterior estimate is obtained by minimizing $\mathcal{J}_i(\theta_i)$, which can be efficiently solved using quasi-Newton methods such as L-BFGS-B (Byrd et al., 1995). We then compute the Hessian of $\mathcal{J}_i$ at $\hat{\theta}_i$, i.e., $\mathbf{H}_i = \nabla_{\theta_i}^2 \mathcal{J}_i(\theta_i)|_{\theta_i=\hat{\theta}_i}$, via finite-difference approximation. A second-order Taylor expansion around $\hat{\theta}_i$ yields the Laplace approximation:

$$p(\theta_i \mid y_i^{t-\kappa:t-1}) \approx \mathcal{N}(\hat{\theta}_i, \mathbf{H}_i^{-1}), \tag{4}$$

where $\mathbf{H}_i^{-1}$ serves as the local covariance estimate of the parameters. We now propagate parameter uncertainty through the nonlinear epidemic dynamics. Let $\Phi$ denote the state-transition operator (i.e. equation 1 for SIR dynamics), $y_i^{t+1} = \Phi(y_i^t, \theta_i)$, where $\mathbf{y}_t^i$ denotes the epidemic state of node $i$ at time $t$. The predictive distribution is formally

$$p(y_i^{t+1}) = \int \Phi(y_i^t, \theta_i)\, p(\theta_i \mid y_i^{t-\kappa:t-1})\, d\theta_i, \tag{5}$$

which is generally intractable. To approximate this integral, we follow the standard deterministic construction used in nonlinear transforms, where a set of $2d + 1$ support points $\{\theta_i^{(m)}\}_{m=0}^{2d}$ with associated weights $\{w^{(m)}\}$ are chosen such that

$$\sum_{m=0}^{2d} w^{(m)} \theta_i^{(m)} = \hat{\theta}_i, \qquad \sum_{m=0}^{2d} w^{(m)} (\theta_i^{(m)} - \hat{\theta}_i)(\theta_i^{(m)} - \hat{\theta}_i)^\top = \mathbf{H}_i^{-1}. \tag{6}$$

In the SIR dynamics considered here, the parameter dimension is $d = 2$, so only $2d + 1 = 5$ support points are required per node. The construction algorithm is provided in Appendix C.2.

**Theorem 3.1** (Second-order accuracy of propagation). *Let $g : \mathbb{R}^d \to \mathbb{R}$ be any twice continuously differentiable function. For a Gaussian random vector $\theta \sim \mathcal{N}(\hat{\theta}, \Sigma)$, the deterministic propagation scheme described above exactly reproduces the first and second moments of $g(\theta)$ up to $\mathcal{O}(\|\theta - \hat{\theta}\|^3)$.*

Theorem 3.1 establishes that the deterministic support point construction above provides an accurate second-order approximation to the predictive mean and variance under nonlinear dynamics. The full proof is provided in Appendix B.1. Propagating each support point through the dynamics yields $y_i^{t+1,(m)} = \Phi(y_i^t, \theta_i^{(m)})$. The predictive mean and uncertainty are then approximated as

$$y_{i,t+1}^{\mathrm{M}} \triangleq \sum_{m=0}^{2d} w^{(m)} y_i^{t+1,(m)}, \qquad u_{i,t+1}^{\mathrm{M}} \triangleq \sum_{m=0}^{2d} w^{(m)} (y_i^{t+1,(m)} - y_{i,t+1}^{\mathrm{M}})^2. \tag{7}$$

## 3.2 DIFFUSION BACKBONE WITH SPATIOTEMPORAL GRAPH TRANSFORMER

In Section 3.1, we provided mechanistic point estimates together with principled uncertainty quantification. However, such estimates have two critical limitations: (i) they cannot capture long-time global spatiotemporal dependencies across the network, and (ii) they are inflexible in accommodating rich contextual information such as mobility patterns, vaccination records, or other exogenous covariates. To overcome these limitations, we introduce a diffusion-based data-driven backbone with a spatiotemporal graph transformer, which learns predictive distributions directly from historical data and integrates diverse contextual signals within a unified generative framework.

During training, we follow the standard denoising diffusion framework (Ho et al., 2020; Song et al., 2020a). To formalize the conditional denoising process, let $\mathbf{Y}^{t-\kappa:t}$ denote the historical observations and $\mathbf{Y}^{t+1:t+h}$ the clean future trajectory. At a sampled diffusion step $s$, the future is corrupted as

$$\mathbf{Y}_s^* = \sqrt{\bar{\alpha}_s}\, \mathbf{Y}^{t+1:t+h} + \sqrt{1 - \bar{\alpha}_s}\, \varepsilon, \qquad \varepsilon \sim \mathcal{N}(0, I), \tag{8}$$

where $\bar{\alpha}_s$ is determined by the noise schedule. To avoid information leakage, we construct two complementary streams: a clean stream $\mathbf{Z}_{\mathrm{clean}} = [\mathbf{Y}^{t-\kappa:t}; \mathbf{0}_{N \times \kappa}]^\top$ and a noisy stream $\mathbf{Z}_{\mathrm{noisy}} = [\mathbf{0}_{N \times h}; \mathbf{Y}_s^*]^\top$. These are concatenated along the temporal axis as $\mathbf{Z} = \mathrm{Concat}(\mathbf{Z}_{\mathrm{clean}}, \mathbf{Z}_{\mathrm{noisy}})$, which is then projected into the latent dimension and fed into the spatiotemporal graph transformer.

The transformer enforces structural priors through a masked attention mechanism. Let $\mathbf{A} \in \{0, 1\}^{N \times N}$ be the adjacency matrix (with self-loops added). We lift $\mathbf{A}$ to the spatiotemporal domain by blockwise repetition, $\tilde{\mathbf{A}} = \mathbf{A} \otimes \mathbf{I}_{2T} \in \{0, 1\}^{(N \cdot 2T) \times (N \cdot 2T)}$, where $T = \kappa + h$ and convert it into an additive bias matrix

$$\mathbf{B}_{ij} = \begin{cases} 0, & \tilde{\mathbf{A}}_{ij} = 1, \\ -\infty, & \tilde{\mathbf{A}}_{ij} = 0, \end{cases} \tag{9}$$

so that attention scores between non-adjacent nodes are suppressed. This guarantees that message passing is restricted to valid spatial edges and self-loops across all temporal slices. The diffusion step $s$ is encoded through a sinusoidal time embedding followed by adaptive layer normalization, providing step-dependent modulation of the hidden states.

After concatenation, the hidden sequence has length $2T$. We apply a linear layer along the temporal dimension that maps $\mathbb{R}^{2T} \to \mathbb{R}^T$, so that the resulting noise estimate $\hat{\varepsilon}$ is temporally aligned with the prediction horizon. Since only the prediction horizon should be supervised, we minimize

$$\mathcal{L} = \|\hat{\varepsilon}^{t+1:t+h} - \varepsilon\|_2^2, \tag{10}$$

where $\hat{\varepsilon}^{t+1:t+h}$ denotes the last $h$ steps of $\hat{\varepsilon}$ aligned with the noisy target $\mathbf{Y}^*$. This objective trains the backbone to denoise corrupted futures while capturing long-range spatiotemporal dependencies under the structural guidance of $\mathcal{G}^{t-\kappa:t}$.

### 3.3 STEERING DIFFUSION INFERENCE WITH UNCERTAIN MECHANISTIC ESTIMATION

While the diffusion backbone provides a strong data-driven framework for capturing global spatiotemporal dependencies and contextual signals, such purely data-driven modeling remains essentially a fit to past history and offers limited guarantees of explainability and robustness. This motivates us to further incorporate mechanistic estimations into the inference process, so that the final forecasts are not only accurate but also epidemiologically consistent.

We incorporate estimates from Section 3.1 as a Gaussian prior distribution directly in the denoising process. Our inference procedure is based on deterministic DDIM sampling (Song et al., 2020a), which enables efficient generation with far fewer steps than ancestral DDPM (Ho et al., 2020). In the standard DDIM update, the clean sample $\mathbf{Y}_0^{t+1:t+h}$ (i.e. the clean prediction for future window $\mathbf{Y}^{t+1:t+h}$) at step $s$ is reconstructed from the noisy state $\mathbf{y}_s$ and the predicted noise $\hat{\varepsilon}_\psi$ (from Section 3.2, $\psi$ is the trained parameters for diffusion backbone) as

$$\mathbf{Y}_0^{t+1:t+h} = \frac{1}{\sqrt{\bar{\alpha}_s}}(\mathbf{Y}_s^{t+1:t+h} - \sqrt{1 - \bar{\alpha}_s}\hat{\varepsilon}_\psi), \qquad (11)$$

where $\bar{\alpha}_s$ is determined by the noise schedule. Here the first term represents the denoised estimate of the clean trajectory, while the second term injects the residual noise predicted by the backbone.

Let $\mathbf{Y}^\mathrm{M} \in \mathbb{R}^{N \times h}$ denote the mechanistic estimation for all $N$ nodes over a forecasting horizon of length $h$. Let $\mathbf{u}^\mathrm{M} \in \mathbb{R}^{N \times h}$ be the corresponding uncertainty matrix, with each entry representing the variance estimate for node $i$ at time $t$ (as quantified in Section 3.1). For notational simplicity, we flatten both matrices into vectors of length $Nh$: $\mathbf{y}_0 := \mathrm{vec}(\mathbf{Y}_0^{t+1:t+h}) \in \mathbb{R}^{Nh}$, $\mathbf{y}^\mathrm{M} := \mathrm{vec}(\mathbf{Y}^\mathrm{M}) \in \mathbb{R}^{Nh}$, and $\mathbf{u}_\mathrm{vec}^\mathrm{M} := \mathrm{vec}(\mathbf{u}^\mathrm{M}) \in \mathbb{R}^{Nh}$. We define the diagonal covariance matrix $\mathbf{\Sigma}_\mathrm{M} = \mathrm{diag}(\mathbf{u}_\mathrm{vec}^\mathrm{M}) \in \mathbb{R}^{Nh \times Nh}$. The reweighted target distribution is then defined as

$$p_\mathrm{target}(\mathbf{y}_0) \propto p_\psi(\mathbf{y}_0 \mid \mathbf{Y}^{t-\kappa:t}, \mathcal{G}^{t-\kappa:t}) \exp\left(-\tfrac{\tau}{2}(\mathbf{y}_0 - \mathbf{y}^\mathrm{M})^\top \mathbf{\Sigma}_\mathrm{M}^{-1}(\mathbf{y}_0 - \mathbf{y}^\mathrm{M})\right), \qquad (12)$$

where the mechanistic prior serves as a reference distribution, and its uncertainty modulates the confidence assigned to each prediction dimension. Here, $\tau$ is a positive constant defined as *guidance scale*, which controls the strength of the mechanistic guidance during inference. Intuitively, our method adds an uncertainty-weighted pull toward the mechanistic estimation at each denoising step. When the mechanistic estimation is confident (small $\mathbf{u}^\mathrm{M}$), the steering aligns the forecast with $\mathbf{Y}^\mathrm{M}$; when uncertainty is large, the prior influence weakens, and the model relies more on spatiotemporal dependencies learned during training. The full inference algorithm is in Appendix C.4.

## 4 EXPERIMENTS

In this section, we comprehensively evaluate `EpiDiff` by answering the following questions:

- **Q1: Performance.** Does `EpiDiff` outperform state-of-the-art epidemic forecasting methods?
- **Q2: Effectiveness.** Does incorporating the mechanistic estimate $\mathbf{y}^\mathrm{M}$ and its associated uncertainty $\mathbf{u}^\mathrm{M}$ improve predictive performance?
- **Q3: Explainability.** How to explain the predictions from `EpiDiff`?
- **Q4: Sensitivity.** How does the performance of `EpiDiff` vary with hyper-parameter settings?

### 4.1 EXPERIMENTAL SETUP

**Datasets.** We evaluate the effectiveness of `EpiDiff` on three representative datasets covering both COVID-19 and influenza-like illness: ***covid-US***, ***covid-JP***, and ***influenza-US***. ***covid-US*** contains state-level daily COVID-19 case counts in the United States from April 15, 2020 to April 15, 2021 (366 days), sourced from The New York Times. ***covid-JP*** comprises prefecture-level daily COVID-19 cases in Japan from April 1, 2020 to September 21, 2021, based on the NHK COVID-19 database. ***influenza-US*** provides weekly state-level influenza incidence in the United States over 158 weeks (2019–2021), estimated from CDC reports. Region-level graphs are constructed according to geographical adjacency following Wang et al. (2022). Data distributions and other details are provided in Appendix D.1.

Table 1: Performance comparison on ***covid-US***, ***covid-JP***, and ***influenza-US*** in RMSE and MAE ($\mathcal{R}$ and $\mathcal{P}$, lower is better). Best results are **in bold** and runner-ups are underlined.

| Methods | *covid-US* | | | | *covid-JP* | | | | *influenza-US* | | | |
|---|---|---|---|---|---|---|---|---|---|---|---|---|
| | $h=7$ | | $h=14$ | | $h=7$ | | $h=14$ | | $h=7$ | | $h=14$ | |
| | $\mathcal{R}\downarrow$ | $\mathcal{P}\downarrow$ | $\mathcal{R}\downarrow$ | $\mathcal{P}\downarrow$ | $\mathcal{R}\downarrow$ | $\mathcal{P}\downarrow$ | $\mathcal{R}\downarrow$ | $\mathcal{P}\downarrow$ | $\mathcal{R}\downarrow$ | $\mathcal{P}\downarrow$ | $\mathcal{R}\downarrow$ | $\mathcal{P}\downarrow$ |
| PatchSIR | $1791_{\pm0}$ | $891_{\pm0}$ | $2077_{\pm0}$ | $886_{\pm0}$ | $313_{\pm0}$ | $98_{\pm0}$ | $332_{\pm0}$ | $106_{\pm0}$ | $323_{\pm0}$ | $82_{\pm0}$ | $454_{\pm0}$ | $138_{\pm0}$ |
| AR | $1791_{\pm16}$ | $835_{\pm16}$ | $1943_{\pm23}$ | $865_{\pm7}$ | $457_{\pm57}$ | $159_{\pm11}$ | $476_{\pm11}$ | $158_{\pm4}$ | $4579_{\pm2429}$ | $2040_{\pm1814}$ | $2328_{\pm236}$ | $1172_{\pm84}$ |
| ARMA | $1832_{\pm29}$ | $843_{\pm2}$ | $1906_{\pm18}$ | $852_{\pm9}$ | $460_{\pm32}$ | $156_{\pm8}$ | $496_{\pm6}$ | $166_{\pm3}$ | $4423_{\pm1971}$ | $2289_{\pm1271}$ | $2314_{\pm150}$ | $1174_{\pm68}$ |
| LSTM | $1797_{\pm59}$ | $867_{\pm48}$ | $1956_{\pm49}$ | $962_{\pm52}$ | $408_{\pm10}$ | $135_{\pm3}$ | $398_{\pm19}$ | $135_{\pm7}$ | $353_{\pm29}$ | $119_{\pm21}$ | $354_{\pm42}$ | $133_{\pm7}$ |
| GRU | $1654_{\pm81}$ | $771_{\pm52}$ | $1949_{\pm38}$ | $974_{\pm30}$ | $364_{\pm12}$ | $120_{\pm3}$ | $383_{\pm8}$ | $130_{\pm3}$ | $377_{\pm41}$ | $142_{\pm13}$ | $498_{\pm303}$ | $192_{\pm75}$ |
| STGCN | $2131_{\pm12}$ | $877_{\pm23}$ | $2446_{\pm51}$ | $1107_{\pm12}$ | $315_{\pm83}$ | $96_{\pm23}$ | $326_{\pm24}$ | $103_{\pm9}$ | $414_{\pm93}$ | $160_{\pm42}$ | $588_{\pm128}$ | $248_{\pm56}$ |
| ColaGNN | $2740_{\pm758}$ | $1175_{\pm339}$ | $3055_{\pm356}$ | $1403_{\pm181}$ | $213_{\pm20}$ | $63_{\pm4}$ | $305_{\pm47}$ | $88_{\pm11}$ | **$250_{\pm17}$** | $85_{\pm9}$ | **$347_{\pm228}$** | $129_{\pm27}$ |
| STGODE | $1763_{\pm96}$ | $719_{\pm34}$ | $2716_{\pm107}$ | $1010_{\pm40}$ | $269_{\pm10}$ | $79_{\pm2}$ | $309_{\pm23}$ | $96_{\pm7}$ | $643_{\pm194}$ | $261_{\pm94}$ | $1415_{\pm460}$ | $570_{\pm200}$ |
| DiffSTG | $1236_{\pm28}$ | $443_{\pm11}$ | $1284_{\pm20}$ | $444_{\pm14}$ | $246_{\pm28}$ | $81_{\pm9}$ | $423_{\pm35}$ | $147_{\pm13}$ | $344_{\pm29}$ | $151_{\pm69}$ | $520_{\pm44}$ | $254_{\pm54}$ |
| STAN | $1364_{\pm196}$ | $1150_{\pm174}$ | $1265_{\pm98}$ | $1046_{\pm90}$ | $222_{\pm87}$ | $127_{\pm40}$ | **$224_{\pm88}$** | $128_{\pm41}$ | $4599_{\pm42}$ | $4306_{\pm32}$ | $3750_{\pm79}$ | $3270_{\pm79}$ |
| EARTH | $2145_{\pm324}$ | $603_{\pm103}$ | $2664_{\pm272}$ | $794_{\pm79}$ | $303_{\pm73}$ | $100_{\pm13}$ | $394_{\pm102}$ | $134_{\pm21}$ | $539_{\pm172}$ | $190_{\pm47}$ | $2027_{\pm1160}$ | $629_{\pm302}$ |
| EpiColaGNN | $1825_{\pm84}$ | $745_{\pm58}$ | $2653_{\pm22}$ | $1235_{\pm9}$ | $282_{\pm33}$ | $83_{\pm11}$ | $327_{\pm19}$ | $99_{\pm6}$ | $1311_{\pm67}$ | $693_{\pm36}$ | $1514_{\pm64}$ | $807_{\pm36}$ |
| EpiDiff | **$1096_{\pm5}$** | **$374_{\pm1}$** | **$1232_{\pm5}$** | **$422_{\pm16}$** | **$188_{\pm7}$** | **$62_{\pm2}$** | $247_{\pm2}$ | **$86_{\pm1}$** | $304_{\pm1}$ | **$82_{\pm0.4}$** | $423_{\pm1}$ | **$125_{\pm0.1}$** |

**Metrics.** The metrics used to evaluate forecasting performance are: root mean squared error ($\mathcal{R}$, RMSE), which measures the square root of the average squared difference between predicted and true values, and mean absolute error ($\mathcal{P}$, MAE), which measures the average absolute difference between predicted and true values. Both $\mathcal{R}$ and $\mathcal{P}$ take values in $[0, +\infty)$, with smaller values indicating better performance. These two evaluation metrics are consistent with previous works (Liu et al., 2023; Wan et al., 2025). Detailed definitions and calculation formulas are provided in Appendix D.2.

**Baselines.** We compare our approach against a diverse set of classic and state-of-the-art forecasting models. Every model is run three times. Detailed descriptions are provided in Appendix D.3.

- *Mechanistic models*: Our propsed mechanistic algorithm **PatchSIR** in Section 3.1 based on **SIR** (Kermack & McKendrick, 1927).

- *Time-series models*: statistical approaches including **Autoregressive (AR)** and **Autoregressive Moving Average (ARMA)** (Contreras et al., 2003), as well as neural sequence models such as **Gated Recurrent Unit (GRU)** (Cho et al., 2014) and **Long Short-Term Memory (LSTM)** (Hochreiter & Schmidhuber, 1997).

- *Graph-based spatiotemporal models*: general-purpose GNN-based models including **STGCN** (Yu et al., 2017) and **ColaGNN** (Deng et al., 2020), a Neural ODE-based model **STGODE** (Fang et al., 2021), and a diffusion-based model **DiffSTG** (Wen et al., 2023). In addition, we include epidemic-aware models that incorporate domain-specific inductive biases: the attention-based model **STAN** (Gao et al., 2021), GNN-based model **EpiColaGNN** (Liu et al., 2023), and a Neural ODE-based model **EARTH** (Wan et al., 2025).

## 4.2 PERFORMANCE

This section addresses **Q1**. To demonstrate the performance of our proposed `EpiDiff`, we conducted comprehensive experiments on various epidemic datasets. We considered multiple baselines as detailed in Tab.1. Key observations include: *(i)* classical time series models (AR, ARMA, LSTM, GRU) perform poorly, especially for non-stationary or long-term epidemic dynamics; *(ii)* spatiotemporal graph models (STGCN, ColaGNN, DiffSTG) and ODE-based methods (STGODE, EARTH) provide moderate improvements but still struggle to adapt to distribution shifts; *(iii)* epidemic-specific neural models (STAN, EpiColaGNN) achieve better robustness, but may lack effective uncertainty modeling; *(iv)* `EpiDiff` consistently outperforms nearly all baselines across datasets and horizons, achieving the lowest or near-lowest RMSE and MAE, and demonstrates strong robustness, especially under distribution shifts such as those seen in ***covid-JP*** and ***influenza-US***. These results highlight the superior accuracy and stability of our approach for real-world epidemic forecasting.

## 4.3 EFFECTIVENESS

The explanation for **Q2** is provided in this section. The first row of Tab.2 shows the results of mechanistic point estimation using the algorithm from Section 3.1. The second row reports the performance of the diffusion backbone with standard DDIM inference, without any steering. The

Table 2: Ablation study of different variants. Best results are **in bold** and runner-ups are underlined.

| Variants | covid-US | | | | covid-JP | | | | influenza-US | | | |
| | $h = 7$ | | $h = 14$ | | $h = 7$ | | $h = 14$ | | $h = 7$ | | $h = 14$ | |
| | $\mathcal{R}\downarrow$ | $\mathcal{P}\downarrow$ | $\mathcal{R}\downarrow$ | $\mathcal{P}\downarrow$ | $\mathcal{R}\downarrow$ | $\mathcal{P}\downarrow$ | $\mathcal{R}\downarrow$ | $\mathcal{P}\downarrow$ | $\mathcal{R}\downarrow$ | $\mathcal{P}\downarrow$ | $\mathcal{R}\downarrow$ | $\mathcal{P}\downarrow$ |
|---|---|---|---|---|---|---|---|---|---|---|---|---|
| w/o Diffusion | 1791 | 891 | 2077 | 886 | 313 | 98 | 332 | 106 | 323 | 82 | 454 | 138 |
| w/o Steering | **1080**±2 | **348**±1 | **1199**±6 | **386**±11 | 370±22 | 94±5 | 417±4 | 121±1 | 438±19 | 224±7 | 1787±87 | 928±59 |
| w/o Uncertainty | 1099±5 | 376±1 | 1253±5 | 438±18 | 316±5 | 97±2 | 365±1 | 122±1 | 328±1 | 147±1 | 446±0.4 | 155±0.3 |
| EpiDiff | 1096±5 | 374±1 | 1232±5 | 422±16 | **188**±7 | **62**±2 | **247**±2 | **86**±1 | **304**±1 | **82**±0.4 | **423**±1 | **125**±0.1 |

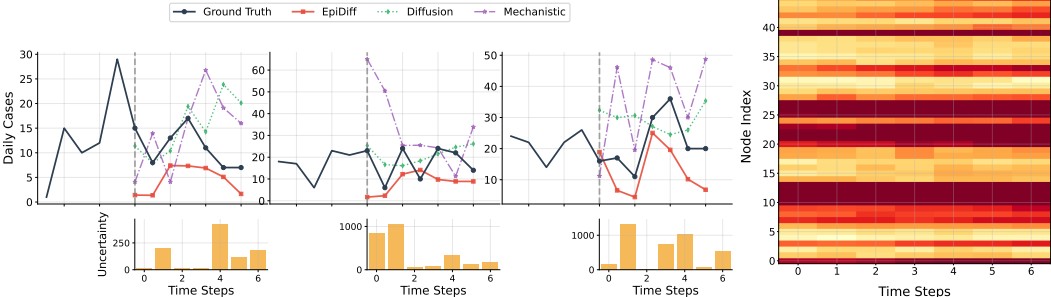

Figure 3: **Left:** Node 30 prediction and uncertainty at windows 30, 60, and 70 on **covid-JP**. **Right:** Uncertainty heatmap for nodes and steps on **covid-JP**. Darker squares indicate greater uncertainty.

third row presents results using only mechanistic point estimate steering, and the final row shows the performance of our full model.

On **covid-US**, a stationary dataset, the accuracy of EpiDiff primarily comes from the strong fitting ability of the diffusion backbone. Since parameter estimation is difficult for non-trending data, uncertainty-steered inference performs slightly worse—but remains very close—to direct diffusion inference. In contrast, on **covid-JP** (positive non-stationarity) and **influenza-US** (negative non-stationarity, both see Appendix D.1), the diffusion backbone alone struggles, while uncertainty-steered inference leads to a substantial improvement, outperforming both the mechanistic model and the diffusion backbone alone.

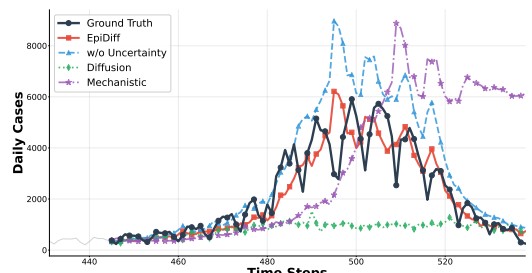

Figure 4: Full test set timeline comparison on Node 12 of the **covid-JP** dataset ($h = 14$).

Fig.4 further illustrates the performance of different variants across the full test set for Node20 in the **covid-JP** dataset. We can see that EpiDiff not only captures the overall trend but also fits multiple outbreak peaks accurately, while other variants fail to track these dynamics as effectively.

## 4.4 EXPLAINABILITY

This section addresses **Q3** on the explainability of EpiDiff. As shown in the left panel of Fig.3, for Node 30 in **covid-JP** at selected windows, the predictive uncertainty bands highlight where the mechanistic model is less confident. In these intervals with high uncertainty, the model's reliance shifts towards the data-driven diffusion backbone, since the mechanistic prior offers weaker guidance. The right panel of Fig.3 visualizes the uncertainty for all nodes and prediction windows as a heatmap. In general, the uncertainty grows with the prediction horizon, while some node–window regions show distinctly high uncertainty. Such visualization helps interpret when and where the model is uncertain, suggesting these high-uncertainty regions as priorities for more fine-grained data collection to further enhance epidemic forecasting.

## 4.5 SENSITIVITY

This section addresses **Q4** by investigating the sensitivity of EpiDiff to the guidance scale parameter $\tau$ in equation 12 (see Fig. 5). As the guidance scale approaches zero, the model behaves like the pure

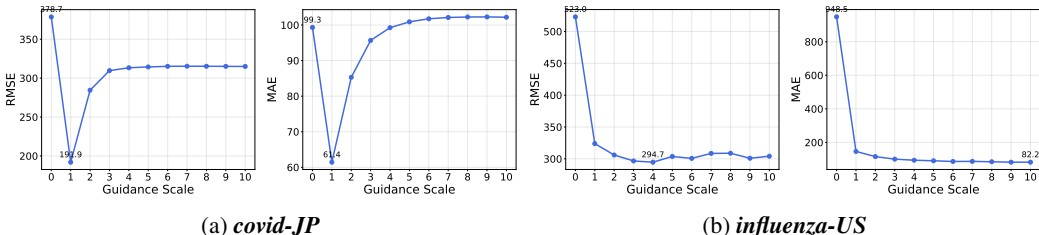

(a) *covid-JP*                    (b) *influenza-US*

Figure 5: RMSE and MAE with respect to guidance scale ($h$=7) on *covid-JP* and *influenza-US*.

diffusion backbone, while larger values push the results toward mechanistic estimation. We observe that the best prediction accuracy is usually achieved at an intermediate guidance scale, highlighting the benefit of combining both data-driven and mechanistic signals. In practice, the optimal scale can be efficiently selected by grid search on the validation set prior to test-time inference.

## 5 RELATED WORKS

### 5.1 MECHANISTIC-INFORMED EPIDEMIC FORECASTING

Traditional epidemic modeling has progressed from purely mechanistic compartmental models (SIR, SEIR) to hybrid frameworks that combine epidemiological domain knowledge with neural networks (Wang et al., 2022; Song et al., 2020b; Tomy et al., 2022; Liu et al., 2023). Early methods mainly used compartmental models to synthesize training data for GNNs in source detection and surveillance (Shah et al., 2020; Ru et al., 2023; Meirom et al., 2021). More recent work explicitly incorporates mechanistic priors into neural architectures—e.g., CausalGNN (Wang et al., 2022) leverages SIRD states as causal features for spatiotemporal learning, and EARTH (Wan et al., 2025) employs SIR-inspired neural ODEs for continuous-time modeling. However, most methods lack explainable fusion mechanisms for mechanistic guidance. In addition, the noise in observations and time-varying parameters render hard-constraint approaches like PINNs (Karniadakis et al., 2021; Raissi et al., 2019) not suited for robust epidemic forecasting. These limitations highlight the need for flexible probabilistic frameworks that can adaptively incorporate mechanistic knowledge and uncertainty.

### 5.2 INFERENCE-TIME STEERING FOR DIFFUSION MODELS

Different from classifier-free guidance (CFG) (Ho & Salimans, 2022), which requires modifying the diffusion model during training to learn conditional and unconditional score functions, inference-time steering methods only intervene during sampling, providing more flexibility. A canonical example is classifier guidance (Dhariwal & Nichol, 2021), which alters the denoising process via gradients from an external classifier. Gradient-based steering extends to generic differentiable reward signals through methods such as universal guidance (Bansal et al., 2023). Particle-based strategies, including best-of-n sampling or Sequential Monte Carlo (SMC) techniques (Wu et al., 2023), further broaden the scope of reward formats, though they may suffer from inefficiency or reduced diversity. Derivative-free approaches like soft value-based decoding (SVDD) (Li et al., 2024) and other sample-resampling mechanisms have been unified under the Feynman–Kac steering framework (Singhal et al., 2025), which formalizes steering as sampling from tilted distributions in a principled way. These advances motivate our use of probabilistic mechanistic priors as structured inference-time guidance for epidemic spatiotemporal forecasting.

## 6 CONCLUSION

In this paper, we introduce `EpiDiff`, a novel hybrid framework for epidemic forecasting that unifies mechanistic models and diffusion-based deep learning. By incorporating mechanistic priors and uncertainty quantification into the diffusion process, `EpiDiff` achieves robust and accurate predictions, even under non-stationary or out-of-distribution scenarios. Extensive experiments on multiple real-world datasets demonstrate the effectiveness and adaptability of our approach. Our work provides practical insights into bridging classical epidemic modeling with modern generative models, paving the way for future advances in epidemiological forecasting and decision-making.

## 7 ETHICS STATEMENT

This work focuses on developing a hybrid diffusion-based framework for epidemic forecasting using publicly available datasets (COVID-19 case counts from the New York Times, NHK COVID-19 database, and CDC influenza reports). These datasets contain only aggregated regional statistics and do not include any personally identifiable information, thereby posing no privacy or security risks.

The proposed methodology is intended to support scientific understanding of epidemic dynamics and to facilitate more reliable epidemic forecasting. We acknowledge that epidemic forecasting can potentially influence public perception and policy decisions; therefore, we emphasize that our framework is designed as a methodological contribution for research purposes rather than as a deployment-ready tool for decision-making.

Our study adheres to the ICLR Code of Ethics. We have no conflicts of interest or external sponsorship that might bias the reported findings. To ensure reproducibility and transparency, we will release code and processed datasets to public upon acceptance.

## 8 REPRODUCIBILITY STATEMENT

We have made significant efforts to ensure the reproducibility of our work. The proposed `EpiDiff` framework is described in detail in Section 3, with the complete algorithmic steps provided in Appendix C. Theoretical results, including the proof of Theorem 3.1 on the accuracy of uncertainty propagation, are given in Appendix B. Detailed descriptions of datasets, preprocessing steps, evaluation metrics, and baselines are included in Appendix D. To support transparency, we provide an anonymous link to our implementation and processed datasets in the paper. All experiments were run with fixed random seeds, and results are reported as averages over multiple runs. Together, these materials ensure that the reported results can be reliably reproduced.

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

## A  USE OF LLMs

We used large language models (LLMs) as general-purpose tools to support the preparation of this work. Specifically, LLMs were employed for polishing the writing of the paper (e.g., improving grammar and clarity of presentation) and for providing assistance with coding tasks such as debugging and formatting. LLMs were not involved in research ideation, experimental design, or the generation of scientific results. All conceptual contributions, methodological designs, analyses, and evaluations were conducted by the authors. The authors take full responsibility for the content of the paper.

## B  THEORETICAL PROOF

### B.1  PROOF OF THEOREM 3.1

*Proof.* Let $\theta \sim \mathcal{N}(\hat{\theta}, \Sigma)$ and denote the centered variable $\delta = \theta - \hat{\theta}$. For any twice continuously differentiable function $g : \mathbb{R}^d \to \mathbb{R}$, we expand $g$ around $\hat{\theta}$ using a second-order Taylor expansion,

$$g(\hat{\theta} + \delta) = g(\hat{\theta}) + \nabla g(\hat{\theta})^\top \delta + \tfrac{1}{2}\, \delta^\top \nabla^2 g(\hat{\theta})\, \delta + R_3(\delta), \tag{13}$$

where the remainder $R_3(\delta)$ is bounded in norm by a constant times $\|\delta\|^3$. Taking expectation under the Gaussian distribution gives

$$\mathbb{E}[g(\theta)] = g(\hat{\theta}) + \tfrac{1}{2}\, \mathrm{tr}\big(\nabla^2 g(\hat{\theta})\, \Sigma\big) + \mathbb{E}[R_3(\delta)]. \tag{14}$$

The nonlinear transform constructs $2d{+}1$ support points $\{\theta^{(m)}\}$ with associated weights $\{w^{(m)}\}$ such that the discrete distribution exactly matches the first two moments of the Gaussian: the weights sum to one, the weighted deviations from $\hat{\theta}$ vanish, and the weighted outer products reproduce $\Sigma$. Propagating these support points through $g$ gives the approximation

$$\widehat{\mathbb{E}}[g(\theta)] = \sum_m w^{(m)} g(\theta^{(m)}). \tag{15}$$

Substituting the Taylor expansion into this expression and using the moment-matching properties shows that the constant and quadratic terms are reproduced exactly, while the linear term cancels. The only discrepancy comes from the remainder terms $R_3(\theta^{(m)} - \hat{\theta})$, which are of order $\|\theta - \hat{\theta}\|^3$. Thus, the estimated mean matches the true mean up to third-order error.

A similar argument applies to the variance. Since $g$ truncated at second order is a quadratic form in $\delta$, and the support points are constructed to match the Gaussian mean and covariance, the variance of the quadratic approximation is captured exactly by the support point rule. The difference between the true variance and the approximate variance again arises only through the cubic remainder, and is therefore of order $\mathcal{O}(\|\theta - \hat{\theta}\|^3)$.

In summary, the nonlinear transform with $2d{+}1$ support points reproduces the first two moments of $g(\theta)$ exactly up to second order, with errors only in higher-order terms. This establishes the second-order accuracy of the propagation scheme. $\square$

## C  DETAILS OF ALGORITHMS

### C.1  NONLINEAR TRANSFORM

The nonlinear transform constructs a set of $2d + 1$ weighted *support points* that deterministically approximate the Gaussian distribution $\mathcal{N}(\hat{\theta}, \Sigma)$. These support points are then propagated through the nonlinear function $g(\cdot)$ to approximate the mean and covariance of $g(\theta)$ up to second order. The procedure is summarized below.

### C.2  NONLINEAR TRANSFORM

The nonlinear transform, also known as the *unscented transform* Julier & Uhlmann (1997); Wan & Van Der Merwe (2000), constructs a deterministic set of $2d{+}1$ weighted *support points* that approximates the Gaussian distribution $\mathcal{N}(\hat{\theta}, \Sigma)$ up to second order. By propagating these support

points through a nonlinear mapping $g(\cdot)$, the first two moments of $g(\theta)$ can be recovered with second-order accuracy, while avoiding the need for explicit numerical integration.

While the nonlinear transform is traditionally employed for nonlinear filtering and state estimation, we adopt it in a different setting. In our framework, the support points are not used to approximate hidden state dynamics, but instead to propagate uncertainty in the mechanistic parameters of epidemic models (e.g., infection and recovery rates). This allows us to quantify how estimation uncertainty in model parameters translates into predictive uncertainty over the epidemic trajectories, and to incorporate this uncertainty directly into the steering of diffusion inference. This application of the nonlinear transform for *uncertainty-aware mechanistic guidance* is, to our knowledge, novel.

---

**Algorithm 1** Nonlinear Transform for Support Point Construction

---

**Require:** Mean $\hat{\theta} \in \mathbb{R}^d$, covariance $\Sigma \in \mathbb{R}^{d \times d}$, scalars $\alpha > 0$, $\nu$; set $\lambda \leftarrow \alpha^2(d + \nu) - d$.
**Ensure:** Support points $\{\theta^{(m)}\}_{m=0}^{2d}$ and weights $\{w^{(m)}\}_{m=0}^{2d}$.
1: Compute the Cholesky factor $L$ of $\Sigma$ such that $\Sigma = LL^{\top}$.
2: $S \leftarrow \sqrt{d + \lambda}\, L$.
3: $\theta^{(0)} \leftarrow \hat{\theta}; \quad w^{(0)} \leftarrow \dfrac{\lambda}{d + \lambda}$.
4: **for** $i = 1$ **to** $d$ **do**
5:     Let $e_i$ be the $i$-th standard basis vector in $\mathbb{R}^d$.
6:     $\theta^{(i)} \leftarrow \hat{\theta} + Se_i; \quad \theta^{(i+d)} \leftarrow \hat{\theta} - Se_i$.
7:     $w^{(i)} \leftarrow \dfrac{1}{2(d+\lambda)}; \quad w^{(i+d)} \leftarrow \dfrac{1}{2(d+\lambda)}$.
8: **end for**
9: **return** $\{\theta^{(m)}, w^{(m)}\}_{m=0}^{2d}$.

---

**Remark.** In the SIR dynamics considered in this paper, the parameter dimension is $d = 2$, so only $2d+1 = 5$ support points are required per node.

### C.3 PATCHSIR

We summarize our method of mechanistic SIR estimation as a robust algorithm **PatchSIR**.

---

**Algorithm 2 PatchSIR**

---

**Require:** History $y_{\text{hist}}$, horizon $T_p$, SIR states $\text{sir}_{\text{hist}}$, priors, dispersion $\alpha_{\text{NB}}$, constraints $\mathcal{B}$
**Ensure:** Forecast $\hat{\mathbf{y}}$, uncertainty $\mathbf{u}$
1: Init normalized SIR states $x_0^{\text{hist}}, x_0^{\text{fore}}$; rolling history $\tilde{y} \leftarrow y_{\text{hist}}$.
2: **for** $t = 1$ **to** $T_p$ **do**
3:     Estimate MAP parameters $\hat{\eta}$, covariance $\Sigma_{\eta}$ (Laplace).
4:     Generate support points $\{(\eta^{(m)}, w^{(m)})\}$ by Nonlinear Transform.
5:     Calibrate scale $s$ from last observed vs. model incidence.
6:     Simulate one-step incidence $z^{(m)}$ for each support point.
7:     **if** samples invalid or extreme **then**
8:         $(\hat{y}_t, u_t) \leftarrow \text{FALLBACKFROMHISTMEAN}(\tilde{y}, 1, \alpha_{\text{NB}})$
9:     **else**
10:        Aggregate with weighted geometric mean $\hat{y}_t$, compute variance $u_t$.
11:     **end if**
12:     Commit $(\hat{y}_t, u_t)$; update history $\tilde{y} \leftarrow \tilde{y} \cup \{\hat{y}_t\}$.
13: **end for**
14: **return** $(\hat{\mathbf{y}}, \mathbf{u})$

---

To ensure reliable estimates and uncertainties even under numerical instabilities, we design the following fallback algorithm.

---

**Algorithm 3** FallbackFromHistMean

---

**Require:** Rolling history $\tilde{y}$, horizon $h = 1$, dispersion $\alpha_{\text{NB}}$
**Ensure:** Fallback forecast $\hat{y}^{\text{fb}}$, uncertainty $u^{\text{fb}}$
  1: $\bar{y} \leftarrow \max(\text{mean}(\tilde{y}), 0)$
  2: $\hat{y}^{\text{fb}} \leftarrow \bar{y}$
  3: $u^{\text{fb}} \leftarrow \bar{y} + \bar{y}^2 / \alpha_{\text{NB}}$
  4: **return** $(\hat{y}^{\text{fb}}, u^{\text{fb}})$

---

## C.4 Steered Diffusion Inference

We present the detailed inference algorithm corresponding to Section 3.1. Our procedure follows the standard deterministic DDIM sampling scheme (Song et al., 2020a), while incorporating an uncertainty-weighted mechanistic prior at each denoising step. Concretely, the clean trajectory $\hat{Y}_0^{t+1:t+h}$ predicted from the noisy state $Y_s^{t+1:t+h}$ is adjusted toward the mechanistic estimate $Y^{\text{M}}$ with strength modulated by the covariance $\Sigma_{\text{M}}$ and guidance scale $\tau$. The full algorithm is given below.

---

**Algorithm 4** Inference Stage of `EpiDiff`

---

**Require:** Trained noise predictor $\hat{\varepsilon}_\psi(\cdot, s)$; schedule $\{\bar{\alpha}_s\}_{s=1}^S$; mechanistic estimate $Y^{\text{M}}$; diagonal covariance $\Sigma_{\text{M}}$; guidance scale $\tau > 0$.
**Ensure:** Forecast $Y_0^{t+1:t+h}$
  1: Initialize $Y_S^{t+1:t+h} \sim \mathcal{N}(0, I)$
  2: **for** $s = S, \dots, 1$ **do**
  3:     $\hat{\varepsilon}_\psi \leftarrow \hat{\varepsilon}_\psi(Y_s^{t+1:t+h}, s)$
  4:     $\hat{Y}_0^{t+1:t+h} \leftarrow \frac{1}{\sqrt{\bar{\alpha}_s}} \left( Y_s^{t+1:t+h} - \sqrt{1 - \bar{\alpha}_s}\, \hat{\varepsilon}_\psi \right)$      ▷ standard DDIM reconstruction
  5:     **Mechanistic steering:** $\hat{Y}_0^{t+1:t+h} \leftarrow \hat{Y}_0^{t+1:t+h} - \tau\, \Sigma_{\text{M}}^{-1} \left( \hat{Y}_0^{t+1:t+h} - Y^{\text{M}} \right)$
  6:     **Re-encode:** $\hat{\varepsilon} \leftarrow \dfrac{Y_s^{t+1:t+h} - \sqrt{\bar{\alpha}_s}\, \hat{Y}_0^{t+1:t+h}}{\sqrt{1 - \bar{\alpha}_s}}$
  7:     **DDIM update:** $Y_{s-1}^{t+1:t+h} \leftarrow \sqrt{\bar{\alpha}_{s-1}}\, \hat{Y}_0^{t+1:t+h} + \sqrt{1 - \bar{\alpha}_{s-1}}\, \hat{\varepsilon}$
  8: **end for**
  9: **return** $Y_0^{t+1:t+h}$

---

# D Details of Experiments

**Infrastructure and Implementation.** All experiments were conducted on a server equipped with NVIDIA A100 GPUs (80GB memory) and an AMD EPYC 7473X CPU with 48 cores and 503GB RAM. Our implementation is based on PyTorch 2.4 with CUDA 12.1.

**Hyperparameter Settings.** For all experiments, we set the historical window length to $T_h = 14$ days. The model was trained with a batch size of 8 for 300 epochs using the Adam optimizer with learning rate 0.002 and weight decay $1 \times 10^{-5}$. Early stopping was applied with a patience of 10 epochs. The diffusion process followed a quadratic noise schedule with $\beta_{\text{end}} = 0.1$ and 200 sampling steps under the DDPM strategy.

Our backbone is the STGTransformer with 4 layers, hidden dimension $d_h = 32$, 4 attention heads, dropout rate 0.2, and channel multipliers $[1, 2]$. Spatial embeddings of size 32 were learned for each nodes. The input is first projected with a $1 \times 1$ convolution, followed by adaptive layer normalization, multi-head attention, and feed-forward networks in each block. The total number of learnable parameters is 75,549. At inference time, we adopt deterministic DDIM sampling with 40 steps, which provides efficient generation while maintaining comparable accuracy to the longer ancestral DDPM trajectories.

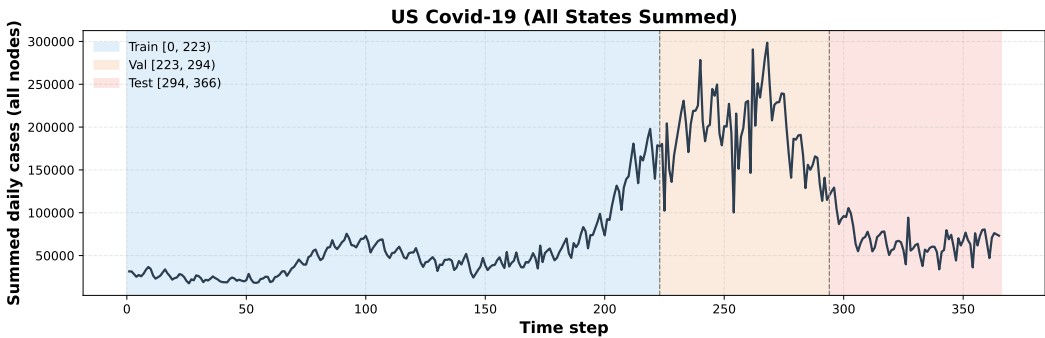

(a) *covid-US* case trends.

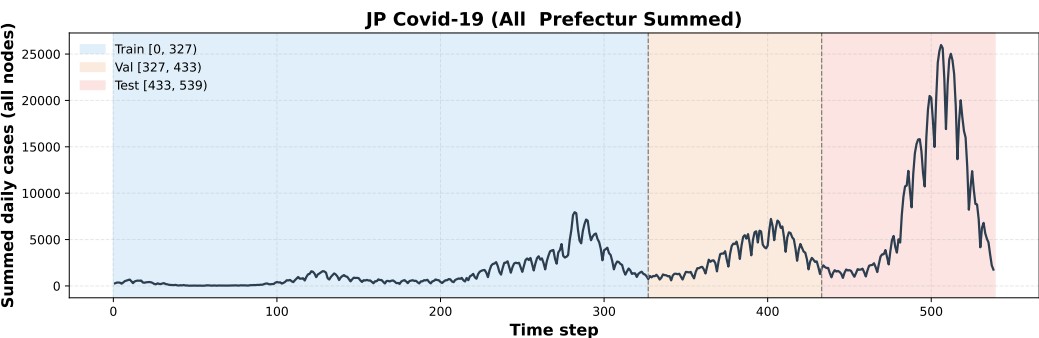

(b) *covid-JP* case trends.

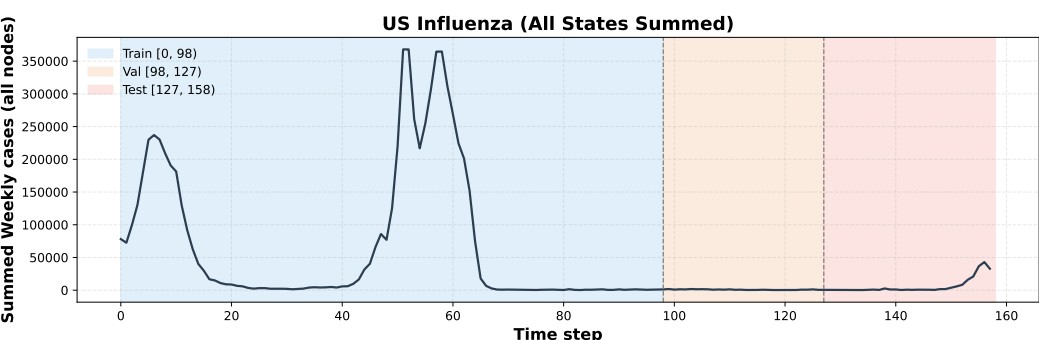

(c) *influenza-US* case trends.

Figure 6: Visualization of case trends in the three datasets: *covid-JP*, *covid-US*, and *influenza-US*.

### D.1 DETAILS OF DATASET

In our experiments, the three datasets represent distinct scenarios commonly encountered in epidemic forecasting. Visualizations of these datasets are shown in Fig. 6. The *covid-US* dataset exhibits a relatively stable distribution, with the test set remaining within the range of the training data. In contrast, *covid-JP* features a significant outbreak in the test period that exceeds anything observed during training, presenting a case of positive distribution shift (non-stationarity). Meanwhile, in *influenza-US*, many influenza cases were reclassified as COVID-19 during the pandemic, resulting in lower estimated values in the test set and thus a negative distribution shift. Through extensive experiments and comprehensive comparisons with state-of-the-art baselines, we demonstrate the effectiveness of the proposed `EpiDiff` framework.

### D.1.1 *covid-US*

Fig. 6a shows the temporal evolution of total daily cases (aggregated over all nodes) in the ***covid-US*** dataset from April 15, 2020 to April 15, 2021. Shaded regions indicate the data splits used in our experiments: **train** `[0, 262)`, **validation** `[262, 313)`, and **test** `[313, 366)`. The data is from daily case report from The New York Times [1].

### D.1.2 *covid-JP*

Trend of daily confirmed COVID-19 cases in ***covid-JP*** is shown in Fig. 6b. Daily cases are aggregated over all prefectures from April 1, 2020 to September 21, 2021. The series is partitioned into **train** `[0, 383)`, **validation** `[383, 460)`, and **test** `[460, 539)`. The data is from the NHK COVID-19 database [2].

### D.1.3 *influenza-US*

Weekly estimated cases are aggregated over all states from Week 1, 2019 to the last week of 2021. The series is partitioned into **train** `[0, 98)`, **validation** `[98, 127)`, and **test** `[127, 158)`.

**Estimation Assumptions.** We derive state-level influenza infections from ILINet reports[3] under the following assumptions: (1) We use the *percent unweighted ILI* as the weekly surveillance indicator, since the weighted version is not consistently available at the state level. (2) State populations are fixed to 2020 U.S. Census estimates and treated as constant over 2019–2021. (3) Each individual seeks outpatient care on average $\mu \approx 3$ times per year ($\mu \approx 0.06$ visits per week) (Cairns & Kang, 2022). (4) Among symptomatic individuals, the probability of seeking medical care is $p_{\text{seek}} \approx 0.45$ (Reed et al., 2009).

**Estimation Formula.** Let $\text{ILI}_{s,t}$ denote the unweighted ILI proportion in state $s$ during week $t$, and let $\text{pop}_s$ denote the population of state $s$. The estimated number of infections is then:

$$\widehat{\text{Infections}}_{s,t} = \frac{\text{ILI}_{s,t} \times \mu \times \text{pop}_s}{p_{\text{seek}}}. \tag{16}$$

**Post-processing.** We round $\widehat{\text{Infections}}_{s,t}$ to the nearest integer, interpolate occasional missing values linearly, and forward-fill at the boundaries.

### D.1.4 POPULATION DATA

U.S. state population data is sourced from the "Annual Estimates of the Resident Population: April 1, 2020 to July 1, 2021 (CO-EST2021-alldata)" published by the U.S. Census Bureau, Population Division (March 24, 2022) [4]. Japanese prefecture population data is obtained from the Wikipedia entry "List of Japanese prefectures by population." [5]

## D.2 DETAILS OF METRICS

We adopt two commonly used metrics to evaluate forecasting performance: root mean squared error ($\mathcal{R}$, RMSE) and mean absolute error ($\mathcal{P}$, MAE). RMSE measures the square root of the average squared deviation between predictions and the ground truth. By squaring the errors, RMSE penalizes large deviations more heavily than small ones. This property makes RMSE particularly sensitive to outliers or sudden spikes in epidemic case counts. In epidemic forecasting, this is useful for highlighting whether a model can capture abrupt outbreak surges. However, the squaring operation may exaggerate the impact of a few extreme errors on the overall evaluation. MAE computes the average absolute deviation between predicted and true values. Unlike RMSE, MAE treats all errors linearly and equally, without disproportionately penalizing large errors. Thus, MAE is more robust to

---

[1] https://github.com/nytimes/covid-19-data

[2] https://github.com/swsoyee/2019-ncov-japan

[3] `https://gis.cdc.gov/grasp/fluview/fluportaldashboard.html`

[4] https://www2.census.gov/programs-surveys/popest/datasets/2020-2021/counties/totals/co-est2021-alldata.csv

[5] https://en.wikipedia.org/wiki/List_of_Japanese_prefectures_by_population

outliers and provides a more balanced assessment of typical prediction accuracy. In epidemic settings, MAE directly reflects the average number of cases by which the model is wrong, making it easier to interpret in practical terms (e.g., "on average, predictions deviate by $x$ cases per day").

Let $\{y_t\}_{t=1}^T$ denote the ground-truth sequence of epidemic values and $\{\hat{y}_t\}_{t=1}^T$ denote the predicted sequence. The metrics are defined as follows:

$$\text{RMSE} = \sqrt{\frac{1}{T}\sum_{t=1}^T (y_t - \hat{y}_t)^2}, \tag{17}$$

$$\text{MAE} = \frac{1}{T}\sum_{t=1}^T |y_t - \hat{y}_t|. \tag{18}$$

Both RMSE and MAE take values in $[0, +\infty)$, with smaller values indicating better predictive performance.

### D.3 DETAILS OF BASELINES

**AR** (Contreras et al., 2003): The Autoregressive (AR) model is a fundamental time series baseline that predicts future values based on a linear combination of past observations. Its simplicity and effectiveness make it a standard benchmark for evaluating more complex forecasting methods.

**ARMA** (Contreras et al., 2003): The Autoregressive Moving Average (ARMA) model combines autoregression with a moving average component, capturing both temporal dependence and short-term noise in time series data. It serves as a widely used baseline for forecasting due to its balance between simplicity and expressive power.

**LSTM** (Hochreiter & Schmidhuber, 1997): The Long Short-Term Memory (LSTM) network is a recurrent neural network architecture designed to capture long-range temporal dependencies by mitigating the vanishing gradient problem.

**GRU** (Cho et al., 2014): The Gated Recurrent Unit (GRU) is a simplified variant of the LSTM that combines the forget and input gates into a single update gate, leading to fewer parameters and faster training while still effectively addressing the vanishing gradient problem.

**STGCN** (Yu et al., 2017): Spatio-Temporal Graph Convolutional Networks (STGCN) is a deep learning framework for traffic forecasting that integrates graph convolution with gated temporal convolution in a fully convolutional architecture. By formulating traffic flow as a graph-structured time series, STGCN effectively captures comprehensive spatial-temporal dependencies while maintaining fast training, fewer parameters, and easier convergence.

**ColaGNN** (Deng et al., 2020): ColaGNN is a cross-location attention based graph neural network designed for long-term influenza-like illness forecasting. It dynamically constructs adjacency matrices using attention scores to capture directed spatial effects, and applies multi-scale dilated convolutions to model both short- and long-term temporal patterns.

**STGODE** (Fang et al., 2021): STGODE is a spatial-temporal forecasting framework that leverages tensor-based ordinary differential equations to jointly capture spatial and temporal dynamics. By enabling deeper architectures, it effectively models long-range dependencies while alleviating the over-smoothing problem of shallow GNNs. STGODE incorporate both semantic adjacency information and dilated temporal convolutions.

**DiffSTG** (Wen et al., 2023): DiffSTG is a probabilistic spatio-temporal graph forecasting framework that generalizes denoising diffusion probabilistic models to STG data. It introduces UGnet, a denoising network that leverages a U-Net architecture for multi-scale temporal dependencies and a GNN for spatial correlations.

**STAN** (Gao et al., 2021): STAN is a spatio-temporal attention network that integrates real-world claims data, demographic similarity, and geographical proximity for pandemic forecasting. It employs a graph attention mechanism to capture complex spatiotemporal disease dynamics and incorporates a dynamics-based loss to improve long-term predictions.

**EpiColaGNN** (Liu et al., 2023): EpiColaGNN is an epidemiology-aware deep learning framework that incorporates the next-generation matrix into both model architecture and objective function. This design enables simultaneous modeling of within-location disease dynamics and cross-location transmission induced by human mobility.

**EARTH** (Wan et al., 2025): EARTH is an epidemiology-aware neural ODE framework designed for epidemic forecasting. It integrates mechanistic compartmental models with deep learning by modeling continuous local transmission dynamics and capturing global infection trends. A cross-attention fusion mechanism further combines global and local signals, enabling more accurate and robust epidemic predictions compared with SOTA methods.

## D.4 COMPARISON OF INFERENCE TIME

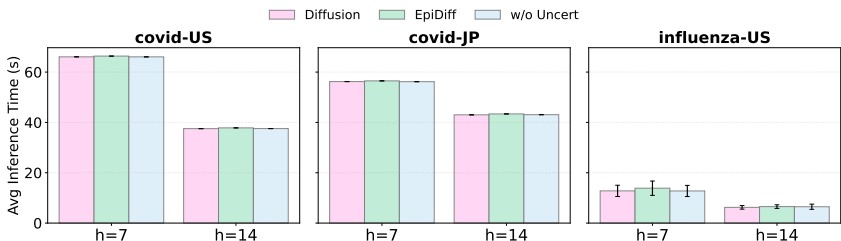

Figure 7: Average inference time (in seconds) for Diffusion baseline, `EpiDiff`, and `w/o` Uncertainty across all datasets and horizons ($h = 7, 14$).

We quantitatively compare the inference time (seconds per sample) of Diffusion, `EpiDiff`, and the w/o Uncertainty variant across three datasets—`covid-US`, `covid-JP`, and `influenza-US`—and two prediction horizons ($h = 7$ and $h = 14$) in Fig.7. All models were evaluated using a 40-step DDIM sampler.

For `covid-US` at $h = 7$, the mean inference times are 66.05 (Diffusion), 66.36 (`EpiDiff`), and 66.04 (w/o Uncert), with standard deviations on the order of $10^{-2}$. At $h = 14$, the corresponding means are 37.53, 37.83, and 37.55. On `covid-JP`, the mean times are 56.23, 56.51, and 56.19 for $h = 7$; 43.01, 43.37, and 43.07 for $h = 14$. For `influenza-US`, the means are 12.80, 13.87, and 12.76 at $h = 7$; 6.24, 6.55, and 6.49 at $h = 14$. In all cases, the standard deviations remain low.

These results show that integrating the mechanistic steering and uncertainty estimation in `EpiDiff` leads to nearly identical inference times compared to the original diffusion baseline and the variant without uncertainty, regardless of the dataset or prediction window. The steering module is plug-and-play and does not noticeably affect inference efficiency when using a 40-step DDIM sampler.

## D.5 SCALABILITY ANALYSIS

**Number of geographies** $N$**.** The total runtime of `EpiDiff` consists of three parts: the time for mechanistic estimation and uncertainty quantification $t_{\text{mech}}$, the time for training the diffusion backbone $t_{\text{train}}$, and the final inference time $t_{\text{inference}}$. Since mechanistic estimation and uncertainty quantification are performed independently for each node and history window, we have $t_{\text{mech}} \propto N$. Consistent with this analysis, Figure 8a and Figure 8b show that both per-epoch training time and overall inference time increase roughly linearly as we subsample more regions. Overall, the runtime scales approximately linearly in the number of geographies.

**Number of Mechanistic Parameters** $n$**.** The additional cost introduced by the mechanistic component depends on the number of parameters in the chosen compartmental model. As discussed in Appendix D.4 (Figure 7), mechanistic steering itself adds almost no overhead to the diffusion backbone, so the dominant factor associated with increasing $n$ is the computation of mechanistic estimation and uncertainty. For a mechanistic model with $n$ parameters, the per-node, per-window cost of our Laplace-based estimator scales with the parameter dimension: approximately $\mathcal{O}(n^2)$ evaluations of the mechanistic objective are required to form the finite-difference Hessian, followed by $\mathcal{O}(n^3)$ time to invert it, on top of the forward simulation cost of the mechanistic model. For

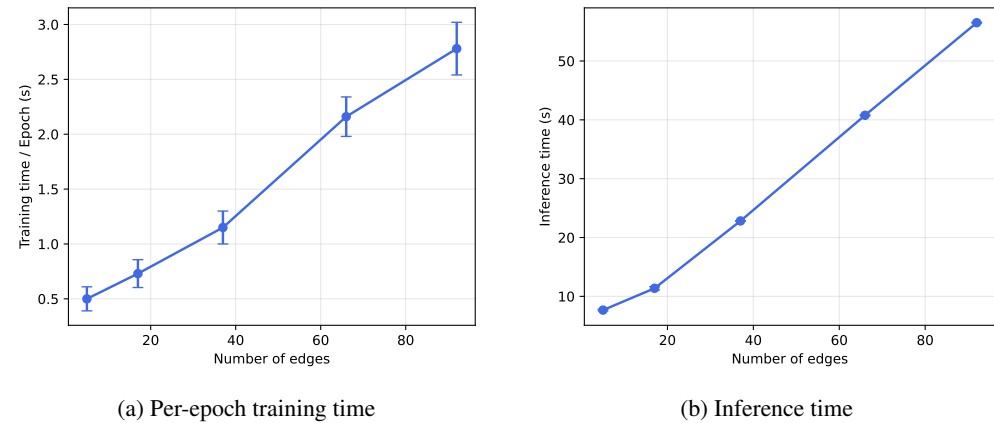

(a) Per-epoch training time         (b) Inference time

Figure 8: On the `covid-JP` dataset, per-epoch training time (left) and overall inference time (right) as we vary the number of nodes by randomly subsampling regions (prediction horizon $h = 7$).

common compartmental families such as SEIR or SIRD, $n$ remains small (typically a few parameters), so the resulting overhead is modest and parallelizable across regions. Overall, the runtime scaling with $n$ is mild, and the diffusion backbone continues to dominate computation on larger graphs.

## D.6 MECHANISTIC ANALYSIS

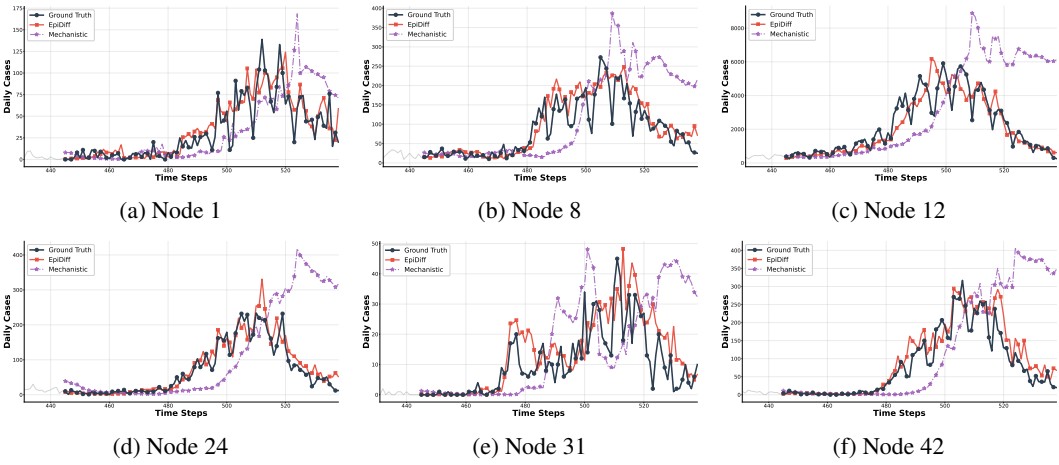

(a) Node 1         (b) Node 8         (c) Node 12

(d) Node 24         (e) Node 31         (f) Node 42

Figure 9: Six representative regions on the `covid-JP` dataset with lookback window $\kappa = 14$ and prediction horizon $h = 14$. For each node, we show daily cases over time for the ground truth, EpiDiff, and the SIR-based mechanistic model.

Model performance is not overly sensitive to the exact accuracy of the mechanistic estimates, because their influence is explicitly modulated by the uncertainty term. Figure 9 shows six representative nodes on the `covid-JP` dataset (lookback window $\kappa = 14$, prediction horizon $h = 14$). in some regions (e.g., Node 1, Node 31) the SIR trajectories are relatively accurate, while in others (e.g., Node 12, Node 42) they deviate substantially and fail to capture local trends. In these regimes, EpiDiff still tracks the ground truth closely: when the mechanistic fit is poor, the posterior uncertainty becomes large, which automatically weakens the guidance and allows the diffusion backbone to correct the mismatch. Thus the mechanistic prior serves as a coarse but informative anchor rather than a brittle constraint, and EpiDiff remains robust to moderate inaccuracies in mechanistic estimation.

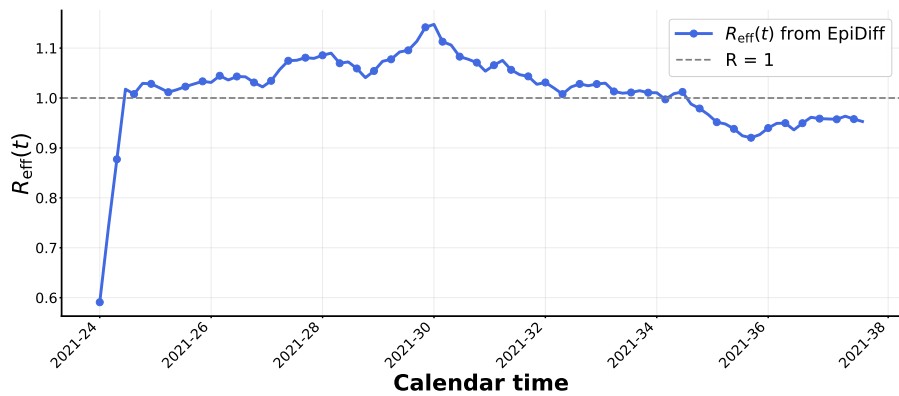

(a) EpiDiff-based $R_{\text{eff}}(t)$ (7-day moving average).

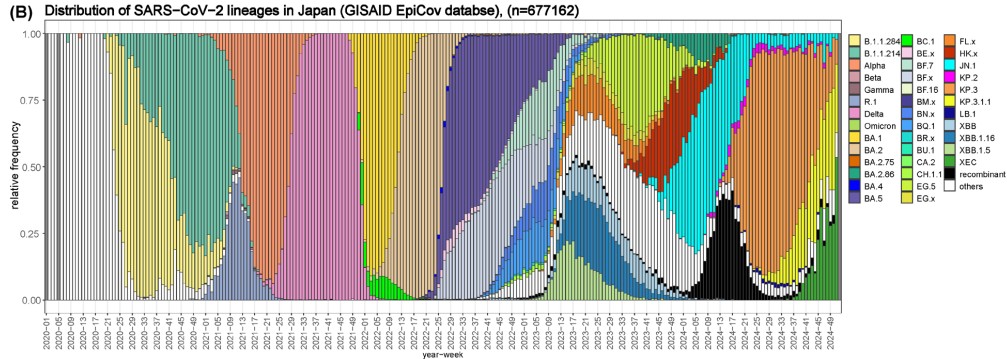

(b) Distribution of SARS-CoV-2 lineages in Japan.

Figure 10: Comparison between the effective reproduction factor estimated by EpiDiff on `covid-JP` (up) and SARS-CoV-2 variant dynamics in Japan (down, reported by Takeuchi et al. (2025)).

### D.7 PRACTICAL VALIDATION

Figure 10a shows the inferred nationwide effective reproduction factor $R_{\text{eff}}(t)$ for `covid-JP`, which remains within a moderate range without pathological spikes: it rises moderately above 1 during growth phases and falls to around or below 1 in declining phases, consistent with typical empirical estimates for COVID-19. Starting from approximately calendar week 25, $R_{\text{eff}}(t)$ exhibits a sharp increase above 1, indicating a period of rapid transmission; this surge coincides with the rapid rise in the proportion of the Delta variant in Japan shown in Figure 10b. Subsequently, $R_{\text{eff}}(t)$ gradually decreases back towards (and below) 1, which is consistent with the phase when Delta becomes dominant and more targeted vaccination efforts and public health interventions are implemented, suggesting that our inferred $R_{\text{eff}}$ captures both variant-driven acceleration and policy-driven mitigation at the national level.

### D.8 SENSITIVITY TO FORECAST HORIZONS AND LOOKBACK WINDOWS

Figure 11 summarizes how the prediction error of `EpiDiff` evolves with the forecast horizon on `covid-JP`. The RMSE increases smoothly as $h$ grows, which is expected as longer-term forecasts accumulate more uncertainty, but there is no sharp degradation. This suggests that the proposed uncertainty-aware guidance helps `EpiDiff` preserve reliable performance even for medium-range (up to 14 days ahead) predictions.

Figure 12 shows how the forecasting error of `EpiDiff` changes with different lookback windows $T_h$ on `covid-JP`. Across $T_h \in \{8, 14, 22, 30, 36\}$, the MAE stays within a narrow range in the high 80s to low 90s, without any sharp degradation. This suggests that the proposed uncertainty-aware

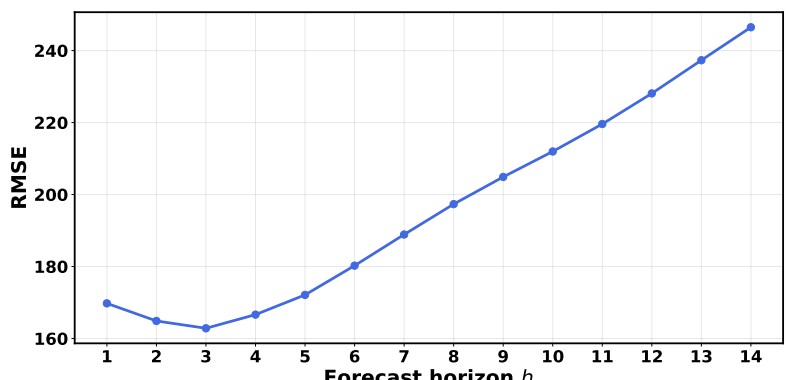

Figure 11: RMSE of `EpiDiff` evolves with the forecast horizon on `covid-JP`.

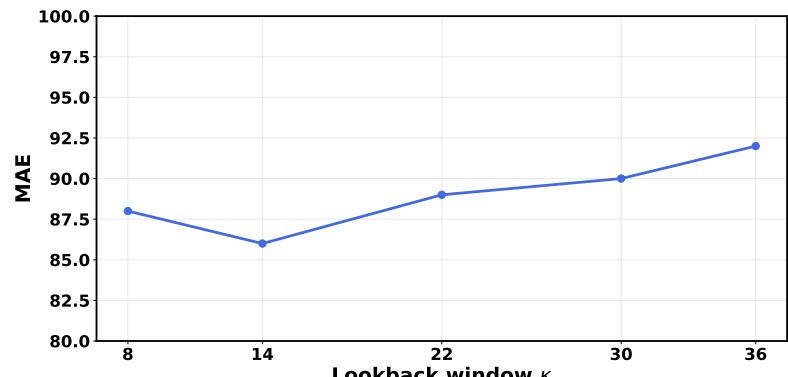

Figure 12: MAE of `EpiDiff` changes with different lookback windows $T_h$ on `covid-JP`.

guidance yields a backbone that is robust to the specific choice of lookback window, rather than relying on a finely tuned history length.

## E  LIMITATIONS

Our approach has several limitations. First, the mechanistic prior is based on simplified SIR-type dynamics with fixed parameters, which may not fully capture real-world heterogeneity or non-stationary interventions. Second, the overall performance is sensitive to the accuracy of mechanistic parameter estimation. As demonstrated in our ablation study, when the estimation is highly noisy or difficult in certain time windows, the resulting uncertainty can propagate into the guidance process and even degrade the predictive performance of the diffusion backbone. Finally, the scalability of our framework to very large graphs remains to be systematically validated. We view these limitations as opportunities for future work, including extensions to richer compartmental models, adaptive and more robust priors, and evaluation at larger scales.

