# OpenReview forum: "Towards Reliable Spatiotemporal Epidemic Forecasting via Steering Diffusion Inference"
_ICLR.cc/2026/Conference — Submitted to ICLR 2026_

### Official Review · Reviewer_xs4R · 2025-10-31

**Soundness:** 2
**Presentation:** 3
**Contribution:** 2
**Rating:** 4
**Confidence:** 4

**Summary:**

This paper introduces EpiDiff, a hybrid framework for epidemic forecasting that integrates mechanistic SIR models with a diffusion-based deep learning model. The key innovation is its ability to quantify the uncertainty of the mechanistic model's predictions. This uncertainty score is then used to dynamically "steer" the diffusion model during inference, allowing the model to rely more on data-driven patterns when the mechanistic model is less confident. Experiments on real-world epidemic datasets show this approach consistently outperforms state-of-the-art baselines in both accuracy and robustness.

**Strengths:**

1. The core contribution is a novel mechanism that uses the posterior uncertainty of a mechanistic model to dynamically steer a diffusion model . This attempts to balance trust between the mechanistic prior and the data-driven backbone, moving beyond simpler hybrid methods that just use model outputs as static features.


2. The empirical evaluation tests the model on three real-world datasets specifically chosen to represent different non-stationary scenarios (stable, positive-shift, and negative-shift). The ablation study (Table 2) effectively isolates the contribution of each model component, demonstrating the framework's performance in these specific cases.


3. The framework offers a specific form of explainability by visualizing the mechanistic uncertainty (Fig. 3). This feature allows a user to identify which of the model's components (the data-driven backbone or the mechanistic prior) is driving the final forecast, offering a view of the model's internal dynamics.

**Weaknesses:**

1. Limited and Incomplete Definition of "Robustness"

The paper's claim of superior "robustness" is based on a limited and incomplete definition. The experiments define robustness almost exclusively as the ability to handle non-stationary distribution shifts, such as the covid-JP outbreak. However, the experiments fail to test for robustness against other critical, real-world data challenges, namely (1) missing values and (2) noisy data.

2. Limited Experimental Scope

The experimental scope is limited. The range of horizons and lookback window sizes studied is narrow, and there is a lack of experiments on larger datasets that feature more nodes and complex temporal patterns.

3. Narrow Uncertainty Quantification and Misleading Explainability

The method's uncertainty quantification is relatively narrow. It only quantifies parameter uncertainty (e.g., $\beta, \gamma$), while ignoring the far greater model uncertainty (that SIR is a flawed model) and data uncertainty (reporting noise). Basing the system's "trust" on this incomplete uncertainty model undermines the central claim of reliability. Consequently, the "explainability" is also misleading; visualizing the model's internal uncertainty is merely a self-referential explanation of its components, not a true epidemiological insight into the causes of an outbreak.

**Questions:**

1. How does the model perform across different forecasting horizons (e.g., 1–8) and lookback window sizes (e.g., 8–36)?
2. Does the model generalize effectively to larger datasets while remaining robust under realistic conditions with missing values and noisy data?
3. How can the model be adapted to account for and explain data uncertainty?
4. Although the experiments claim to address non-stationary settings, there is no quantification of the data’s distribution shift. How does the model perform under varying degrees of distribution shift?

---

> ### Author Response · Authors · 2025-11-20
>
> We sincerely appreciate your efforts to review our paper and provide valuable suggestions. Below we address each concern in detail.
>
> ---
> - w1: Limited and Incomplete Definition of "Robustness".
> - **R w1**: We agree that robustness is a multi-faceted concept and that missing values and noisy observations are important real-world challenges. In this work, our use of **"robustness" is explicitly limited to robustness against non-stationary epidemic regimes and distribution shifts as stated in Line 70-75**, which are the main failure modes we target (for example, new waves, as in covid-JP). Handling missing and noisy data is largely orthogonal to the proposed guidance mechanism: we handle missing values by filling them with a moving average during preprocessing, while noise is inherent in the data, and spatiotemporal forecasting tasks typically assume that observations are reliable enough to be used as inputs. A more systematic study of robustness to missingness and observation noise, possibly by combining EpiDiff with stronger imputation or denoising modules, is a complementary direction that we plan to explore in future work.
>
>
> ---
> - w2: Limited Experimental Scope.
> - **R w2**: We respectfully note that our experimental design is consistent with prior work and remains practically meaningful. In [1], the lookback window is fixed to 20 and the prediction horizons are 5 and 10, on three datasets (Australia-COVID, US-Regions, US-States) with 8, 10, and 49 nodes, respectively. In [2], only horizons 1, 2, and 4 are reported, and the evaluation dataset US-HHS-Flu has 10 nodes. In comparison, our current setup is already sufficient to validate our claims and findings.
>
> In addition, we have strengthened the analysis in the revised version. At line 1118, Appendix D.5 now provides a complexity analysis with respect to both the number of geographies and the number of mechanistic parameters, showing that runtime grows approximately linearly with the number of nodes, which we further validate by subsampling nodes. Finally, at line 1232, we add experiments varying the lookback windows and forecast horizon, demonstrating that EpiDiff maintains stable performance under different temporal settings.
>
>
> ---
> - w3: Narrow Uncertainty Quantification and Misleading Explainability.
> - **R w3**: We respectfully disagree that our uncertainty and explainability are misleading. As stated in lines 70–75 and illustrated in Figure 1 of the main paper, our aim is to address a specific gap in current hybrid epidemic forecasting models: **how to combine a mechanistic prior with a data-driven backbone in a reliability-aware way**. Within this scope, we focus on **mechanistic parameter uncertainty** as a tractable signal that modulates the influence of the SIR prior and makes explicit when the system leans more on mechanistic structure versus learned patterns.
>
> We agree that model misspecification and data noise are important related to uncertainty, but they **fall outside the problem setting we explicitly define**. Our notion of reliability is therefore correspondingly scoped: when the SIR fit is locally weak (high posterior variance), its impact on the forecast is reduced rather than treated as uniformly trustworthy. Likewise, our notion of explainability is not to provide causal epidemiological insight into the origins of an outbreak, but to **make transparent how the forecasting system internally balances mechanistic and data-driven components**.

---

> > ### Author Response · Authors · 2025-11-20
> >
> > - q3: Adaptation for data uncertainty.
> > - **R q3**: In this work, we do not explicitly model **data uncertainty**: our starting point, consistent with most spatiotemporal epidemic forecasting models, is that after basic preprocessing (e.g., moving-average imputation for missing values), the reported observations are treated as sufficiently reliable inputs, while the noise in case counts is inherent to the surveillance process. Quantifying and explaining data uncertainty is therefore **not the direct target of EpiDiff**, but rather a separate layer that can be handled by complementary methods. Recent work such as [3] uses additional mobility data and a specified mechanistic model to **calibrate** raw observations, while nowcasting methods like [4] explicitly model reporting delays and under-reporting to recover more reliable current incidence. These approaches are complementary to EpiDiff: they aim to produce cleaner, uncertainty-aware observations, whereas EpiDiff focuses on **reliability-aware forecasting given those observations**, by modulating mechanistic guidance with parameter uncertainty. In future extensions, one could combine these two layers by feeding calibrated or nowcasted incidence (and its uncertainty) into our guidance term, but a full treatment of data uncertainty lies outside the scope of the present paper.
> >
> >
> > ---
> > - q4: Performance under varying degrees of distribution shift.
> > - **R q4**: A simple and interpretable way to characterize non-stationarity in our setting is to compare the mean level of the training and test periods, e.g., by $\Delta\mu = \text{mean(test)} - \text{mean(train)}$ for the aggregated case counts. As illustrated in Figure 6, the three datasets we use already span different degrees of non-stationarity (from relatively mild shifts to pronounced regime changes), and Table 1 shows that EpiDiff performs consistently well across these distinct regimes. This indicates that the model is robust under varying levels of distribution shift.
> >
> > ---
> > [1] Wan, Guancheng, et al. "EARTH: Epidemiology-Aware Neural ODE with Continuous Disease Transmission Graph." Forty-second International Conference on Machine Learning.
> >
> > [2] Liu, Mutong, Yang Liu, and Jiming Liu. "Epidemiology-aware deep learning for infectious disease dynamics prediction." Proceedings of the 32nd ACM International Conference on Information and Knowledge Management. 2023.
> >
> > [3] Chang, Serina, et al. "Mobility network models of COVID-19 explain inequities and inform reopening." Nature 589.7840 (2021): 82-87.
> >
> > [4] Sahai, Saumya Yashmohini, et al. "A machine learning model for nowcasting epidemic incidence." Mathematical Biosciences 343 (2022): 108677.

---

> ### Author Response · Authors · 2025-11-26
>
> Dear Reviewer,
>
> Thank you again for your time and constructive comments.
>
> As we **enter the final week of the discussion period**, we wanted to follow up to ensure that our response has adequately addressed your concerns. We would greatly appreciate your feedback, as your evaluation is crucial to us.
>
> We remain available to answer any further questions you may have.
>
> Best regards,
>
> The authors of Submission 9683

---

### Official Review · Reviewer_tsgY · 2025-11-01

**Soundness:** 3
**Presentation:** 3
**Contribution:** 3
**Rating:** 6
**Confidence:** 3

**Summary:**

The paper proposes EpiDiff, a hybrid epidemic forecasting framework that combines diffusion-based deep learning with mechanistic SIR modeling. By using uncertainty-aware steering to guide the diffusion process, EpiDiff achieves accurate and robust forecasts while maintaining epidemiological consistency, outperforming prior neural and hybrid methods on COVID-19 and influenza datasets.

**Strengths:**

1. A motivation that closely aligns with real-world epidemic applications.
2. Clear and easy-to-follow writing.
3. Comprehensive theoretical analysis provides strong support for the proposed method.

**Weaknesses:**

1. Why does introducing the SIRS prior help address the OOD problem? The data fitted by SIRS are still IID, and SIRS itself does not inherently possess generalization capability.
2. Regarding the claim in line 241, how does the diffusion backbone integrate diverse contextual signals such as mobility patterns, vaccination records, or other exogenous covariates?
3. In Table 2, why does removing Steering improve performance on the COVID-US dataset? COVID-US is a stationary dataset, but that does not necessarily mean it is non-trending; parameter estimation should still be feasible.
4. In Figure 3, the right-hand plot suggests that uncertainty appears to be node-dependent rather than time-dependent. Does this imply that certain nodes are inherently harder to predict?

**Questions:**

see weakness

---

> ### Author Response · Authors · 2025-11-20
>
> We sincerely appreciate your efforts to review our paper and provide valuable suggestions. Below we address each concern in detail.
>
> ---
> - w1: Why does introducing the SIR prior help address the OOD problem?
> - **R w1**: Introducing the SIR prior helps under OOD conditions because it **captures basic epidemic regularities** (such as bounded growth and eventual saturation) that remain valid even when the observed data distribution shifts across waves or interventions. A purely data-driven diffusion backbone must extrapolate from past patterns and can behave unpredictably when those patterns change, whereas the SIR prior provides a **structurally plausible reference trajectory** that gently pulls the backbone toward epidemiologically consistent forecasts when the posterior uncertainty is low. At the same time, our uncertainty estimate naturally limits this effect over longer horizons: as the prediction window grows, the mechanistic uncertainty increases and the **effective guidance strength decays**, so the **backbone gradually dominates for long-range forecasts** while still benefiting from the prior in the near term.
>
>
> ---
> - w2: How does the diffusion backbone integrate diverse contextual signals?
> - **R w2**: We apologize for the confusion. In the current version of EpiDiff, the diffusion backbone primarily integrates **regional graph structure** through the bias matrix in Eq. (9), which encodes adjacency information into the attention scores. We see two natural extensions of our architecture to incorporate multi-modal signals. First, the **bias matrix in Eq. (9)** can be generalized to include additional context, for example by defining context-aware pairwise terms so that the attention between two nodes depends not only on the graph but also on covariates or metadata. Second, inspired by classifier-free guidance, one can **add embeddings of auxiliary signals** (such as mobility, interventions, or wastewater) to the transformer representations before noise estimation, so that the backbone conditions directly on these modalities. We consider systematically integrating such multi-modal contextual signals for both forecasting and downstream decision making as an important direction for future work.
>
>
> ---
> - w3: Why does removing Steering improve performance on the COVID-US dataset?
> - **R w3**: On COVID-US, removing steering slightly improves performance because **the diffusion backbone already models this in-distribution dataset very well**, while the SIR prior is only weakly informative. Although COVID-US is approximately stationary, it still exhibits multi-wave patterns and reporting artifacts that a simple single-patch SIR model cannot fully capture, even when parameter estimation is feasible. In this setting, adding steering can introduce a **small bias** when the mechanistic trajectory does not perfectly align with the observed dynamics, so the backbone-only variant performs marginally better. This does not contradict our overall claim: on other datasets where **distribution shift is stronger or the mechanistic signal is more informative**, steering consistently helps, and on covid-US the uncertainty-aware design already limits the impact of a slightly misaligned prior.
>
>
> ---
> - w4: Interpretability of right-hand plot in Figure 3.
> - **R w4**: This is an insightful observation. The right-hand plot in Figure 3 indeed shows a clear **node-dependent pattern** in mechanistic uncertainty, indicating that the SIR prior is systematically less reliable for some nodes than for others (for example, due to noisier reporting or more irregular local dynamics), rather than that these nodes are inherently "unpredictable." At the same time, the pattern is not purely node-specific: along the time axis, uncertainty **tends to increase for larger prediction steps**, which is expected because the mechanistic forecast becomes less certain as the horizon grows. In EpiDiff we interpret these high-uncertainty regions as precisely where the mechanistic prior should be trusted less, so the model relies more on the diffusion backbone there, while shorter horizons and more regular nodes benefit more from the prior. This behavior is consistent with our design of uncertainty-aware guidance.

---

> ### Author Response · Authors · 2025-11-26
>
> Dear Reviewer,
>
> Thank you again for your time and constructive comments.
>
> As we **enter the final week of the discussion period**, we wanted to follow up to ensure that our response has adequately addressed your concerns. We would greatly appreciate your feedback, as your evaluation is crucial to us.
>
> We remain available to answer any further questions you may have.
>
> Best regards,
>
> The authors of Submission 9683

---

### Official Review · Reviewer_YpmH · 2025-11-01

**Soundness:** 2
**Presentation:** 4
**Contribution:** 1
**Rating:** 2
**Confidence:** 4

**Summary:**

This paper presents a diffusion model that incorporates an SIR model as a classifier-type guidance for improving time series forecasting in epidemics. This approach is tested in a few datasets.

**Strengths:**

* The motivation and underlying technical ideas are reasonable and do not appear to contain major errors.

* Good presentation and visualizations.

**Weaknesses:**

* This paper addresses a spatiotemporal setting in which time series are predicted for multiple locations. However, the SIR model used in the study is defined for a single location. It is well established that when modeling spatiotemporal dynamics, the SIR framework must incorporate connections between locations to capture the exchange of cases across them (see, for example, [1]). In addition, it is well known that the SIR model is not suitable for modeling COVID-19 dynamics, see [4] for a review on more appropriate models. Therefore, the mechanistic model employed in this paper is not appropriate for the spatiotemporal setting considered.

* The methodology section includes an extensive discussion on uncertainty quantification, yet the experiments do not report a single metric evaluating uncertainty calibration. The paper’s motivation highlights the “lack of sufficient flexibility to handle uncertainty,” but it remains unclear why the authors chose not to present any uncertainty analysis. This omission is particularly notable given that several recent works on diffusion models for forecasting explicitly emphasize and evaluate uncertainty calibration (see, for example, [2]), whereas this paper appears to disregard it entirely.

* The mechanistic guidance is not a novel technical contribution, despite the authors’ emphasis, as similar ideas have already been explored in several prior works (e.g., [3]).

* While the authors claim that the mechanistic model enhances explainability, no experiments are presented to demonstrate or validate this claim.

[1] Lloyd, A.L. and Jansen, V.A., 2004. Spatiotemporal dynamics of epidemics: synchrony in metapopulation models. Mathematical biosciences, 188(1-2), pp.1-16.

[2] Rühling Cachay, S., Zhao, B., Joren, H. and Yu, R., 2023. Dyffusion: A dynamics-informed diffusion model for spatiotemporal forecasting. Advances in neural information processing systems, 36, pp.45259-45287.

[3] Huang, J., Yang, G., Wang, Z. and Park, J.J., 2024. DiffusionPDE: Generative PDE-solving under partial observation. Advances in Neural Information Processing Systems, 37, pp.130291-130323.

[4] Adiga, A., Dubhashi, D., Lewis, B., Marathe, M., Venkatramanan, S. and Vullikanti, A., 2020. Mathematical models for covid-19 pandemic: a comparative analysis. Journal of the Indian Institute of Science, 100(4), pp.793-807.

**Questions:**

How do you account for the fact that different regions (e.g., U.S. states) vary greatly in scale? For instance, California, being a populous state, will naturally have much higher disease incidence than a smaller state like Arkansas. Do you apply any normalization or weighting to ensure that your evaluation is not biased toward performance in large regions while overlooking smaller ones?

---

> ### Author Response · Authors · 2025-11-20
>
> We sincerely appreciate your efforts to review our paper and provide valuable suggestions. Below we address each concern in detail.
>
> ---
> - w1: SIR is not approriate to model COVID-19 dynamics.
> - **R w1**: We appreciate the reviewer’s concern. Our choice of simple SIR model for mechanistic modeling is primarily driven by **computational efficiency** and is supported by **strong empirical evidence from prior work**. (1) As noted in line 223, the SIR model contains only two parameters $\beta, \gamma$, so the required Hessian is merely $2 \times 2$, allowing us to compute uncertainty with only five support points, which makes the mechanistic steering **more efficient**; (2) Importantly, SIR-based models have been shown to perform competitively for COVID-19 forecasting. For example, [1] reports in its Table 2 that **a single-patch SIR model achieves lower MAE and MAPE than a network-based SEIR model for US state-level and county-level COVID-19 predictions**. In addition, [2] provides real-world evidence that early-stage COVID-19 dynamics can be accurately captured by classical SIR models, which serve as reliable baselines before more complex compartmental models become identifiable; (3) Moreover, our framework does not rely on SIR-specific assumptions. The mechanistic prior is fully modular and can be replaced by **any** compartmental model (e.g., SEIR, SIRD).
>
> ---
> - w2: Missing uncertainty analysis.
> - **R w2**: The uncertainty quantification we discuss refers to **the relative uncertainty induced by estimating the SIR model parameters**, expressed across different regions and time steps and we have analyzed both the validity and sensitivity of this uncertainty measure in the experimental section. (1) First, Definition 2 in Section 3.1 specifies the meaning of uncertainty in our context, and Eq. (12) introduces the guidance scale $\tau$ that governs its influence during inference. (2) Second, we provide extensive experiments on our defined mechanistic uncertainty. **Table 2** reports an ablation study that isolates the contribution of uncertainty-guided steering. **Figure 3** visualizes the resulting spatiotemporal uncertainty patterns. **Figure 5** evaluates the sensitivity of the method to different guidance scales. (3) Third, EpiDiff performs DDIM-style deterministic inference and produces standard spatiotemporal point predictions following prior works such as [1, 3]. The predictive variability of EpiDiff is reflected through the error bars reported in **Table 1**.
>
> ---
> - w3: The mechanistic guidance is not a novel technical contribution.
> - **R w3**: The novelty of EpiDiff lies in **applying guidance within a noisy and stochastic epidemic dynamic system**, where we **jointly use a mechanistic estimate and its quantified posterior uncertainty** to construct a Gaussian prior for probabilistic steering. Prior works such as [4] focus on **deterministic PDE settings**, where guidance is implemented through **PDE-residual corrections** to enforce physical constraints during sampling. These approaches operate under fundamentally different settings and do not incorporate uncertainty quantification for steering a stochastic forecasting process.
>
> ---
> - w4: Lack validatation of model explainability.
> - **R w4**: We would like to clarify that the form of *explainability* targeted by EpiDiff is to **make explicit how the model’s reliance shifts between the mechanistic prior and the data-driven backbone across different spatiotemporal regimes**. This is directly supported by the experiments already included in the paper. In Figure 3 (Section 4.4), the left panel visualizes, for a representative node on covid-JP, the mechanistic prediction together with its predictive uncertainty bands over selected windows, highlighting intervals where the mechanistic model is less confident and therefore provides weaker guidance to the diffusion backbone. The right panel summarizes this behavior as a heatmap of mechanistic uncertainty across all nodes and prediction windows, revealing when and where the model is intrinsically uncertain and thus expected to lean more on the data-driven component. These analyses collectively substantiate our explainability claim by making the internal balance between mechanistic guidance and learned dynamics observable in a measurable and reproducible way.

---

> > ### Author Response · Authors · 2025-11-20
> >
> > - q1: How to scale variability in regional infectious population.
> > - **R q1**: We **normalize each region’s time series to avoid bias toward large-population nodes**. Following prior work such as [5], we compute the per-node mean and standard deviation on the training set and use these statistics to normalize all inputs. During guidance, the mechanistic estimate $y_M$ is normalized using the same per-node statistics so that both the backbone and the mechanistic prior operate on a comparable scale. After forecasting, we denormalize the outputs back to the original units for evaluation. This procedure prevents large regions from dominating the training objective and ensures that smaller regions are treated fairly.
> >
> > ---
> > [1] Wang, Lijing, et al. "Causalgnn: Causal-based graph neural networks for spatio-temporal epidemic forecasting." Proceedings of the AAAI conference on artificial intelligence. Vol. 36. No. 11. 2022.
> >
> > [2] Liu, Taoran, et al. "A real-world data validation of the value of early-stage SIR modelling to public health." Scientific Reports 13.1 (2023): 9164.
> >
> > [3] Wan, Guancheng, et al. "EARTH: Epidemiology-Aware Neural ODE with Continuous Disease Transmission Graph." Forty-second International Conference on Machine Learning.
> >
> > [4] Huang, Jiahe, et al. "DiffusionPDE: Generative PDE-solving under partial observation." Advances in Neural Information Processing Systems 37 (2024): 130291-130323.
> >
> > [5] Wen, Haomin, et al. "Diffstg: Probabilistic spatio-temporal graph forecasting with denoising diffusion models." Proceedings of the 31st ACM international conference on advances in geographic information systems. 2023.

---

> ### Author Response · Authors · 2025-11-26
>
> Dear Reviewer,
>
> Thank you again for your time and constructive comments.
>
> As we **enter the final week of the discussion period**, we wanted to follow up to ensure that our response has adequately addressed your concerns. We would greatly appreciate your feedback, as your evaluation is crucial to us.
>
> We remain available to answer any further questions you may have.
>
> Best regards,
>
> The authors of Submission 9683

---

### Official Review · Reviewer_iLWB · 2025-11-01

**Soundness:** 3
**Presentation:** 3
**Contribution:** 3
**Rating:** 2
**Confidence:** 2

**Summary:**

This paper proposes EpiDiff, a hybrid spatiotemporal epidemic forecasting framework that combines mechanistic models with diffusion-based neural networks. The authors proposed a novel uncertainty-aware mechanistic guidance (via Laplace approximation and nonlinear transforms) for steering the inference process (at evaluation phase) of the predictive diffusion model. The method demonstrates strong performance on COVID-19 and influenza datasets, especially under distribution shifts.

**Strengths:**

The paper is well-written, ideas are well-presented and easy to follow.
The idea of using Laplace approximation for the posterior distribution and propagates the parameter uncertainty to predictive uncertainty via support points is novel and efficient.
Comprehensive experiments and ablation studies showcasing the efficacy of the proposed components.

**Weaknesses:**

The parameters $\hat{\theta}_i$ are estimated separately for each history window $y^{t−\kappa:t−1}$ which raises concerns about the consistency of the mechanistic model over one epidemic season. Also the prior distribution $p(θ_i)$ is not specified.
    The estimated parameters for the mechanistic model are not shown, which limits the interpretability of the model.
    As shown in the ablation study, model performance is sensitive to guidance scale $\tau$, which limits the application of the model in real-world setting.
    The model is a direct combination of previous works ([1] and [2]), the technical novelty is somewhat limited.


[1] Wen, Haomin, et al. "Diffstg: Probabilistic spatio-temporal graph forecasting with denoising diffusion models." Proceedings of the 31st ACM international conference on advances in geographic information systems. 2023.
[2] Singhal, Raghav, et al. "A general framework for inference-time scaling and steering of diffusion models." arXiv preprint arXiv:2501.06848 (2025).

**Questions:**

How the estimated parameters of mechanistic model change over time? And how did you choose the prior?
    How does the framework perform when the mechanistic model is mis-specified, which is a common scenario for real-world data with small sample size?

---

> ### Author Response · Authors · 2025-11-20
>
> We sincerely appreciate your efforts to review our paper and provide valuable suggestions. Below we address each concern in detail.
>
> ---
> - w1: Consistency and specification of mechanistic parameter estimates.
> - **R w1**: We agree that estimating parameters completely independently in each history window could in principle lead to inconsistencies over an epidemic season. In our implementation, however, the mechanistic parameters are not re-estimated from scratch each time. For window $k$, we initialize the optimization at the estimate obtained for window $k-1$, so the new estimate is obtained by locally updating a previous solution rather than fitting an entirely separate model. Conceptually, this corresponds to using the previous estimate as an informative prior for the current window, which encourages temporal consistency of the mechanistic parameters while still allowing them to adapt to non-stationarities such as interventions or behavioral changes.
>
> Since the SIR component is used only as a **coarse-grained prior** rather than a fully calibrated epidemiological model, its fitted parameters are not sufficiently reliable to be directly interpreted or reported. Instead, we derive aggregate quantities (such as nationwide $R_{\mathrm{eff}}(t)$) from the final EpiDiff predictions. In Appendix D.7, we further provide a **practical validation** of EpiDiff by comparing the inferred nationwide $R_{\mathrm{eff}}(t)$ with the variant dynamics in Figure 10: **$R_{\mathrm{eff}}$ rises sharply from around week 25 of 2021, aligning with the rapid increase of the Delta variant in Japan, and then declines as vaccination and public health interventions take effect**, confirming that our estimates are both numerically reasonable and epidemiologically consistent.
>
> ---
> - w2: Performance is sensitivity to guidance scale $\tau$.
> - **R w2**: Our ablation on the guidance scale $\tau$ indeed shows that it affects performance, but **we do not observe prohibitive sensitivity that would prevent practical use**. In particular, the results **are stable over a reasonably wide range of $\tau$**; for example, on both covid-JP and influenza-US the accuracy remains strong when $\tau \in (1, 2)$, and performance only degrades when $\tau$ is set to extreme values that either almost ignore the mechanistic prior or almost completely override the data-driven backbone, which is the expected behavior for a guidance weight. In practice, **$\tau$ can be selected once per dataset on a validation set**, similar to other standard hyperparameters such as the learning rate. Moreover, in EpiDiff the effective strength of guidance is further modulated by the estimated mechanistic uncertainty $u_M$, so the model already downweights the prior when it is unreliable.
>
> ---
> - w3: EpiDiff is a direct combination of Diffstg [1] and FK Steering [2].
> - **R w3**: Although EpiDiff is motivated by Diffstg and Feynman–Kac (FK) steering as cited in introduction and related works part, **both our diffusion backbone design and our inference-time steering mechanism are novel**. (1) DiffSTG was designed for traffic flow prediction, where the noise predictor uses a GCN+TCN U-Net. In contrast, EpiDiff targets spatiotemporal epidemic forecasting, and in the diffusion training stage we introduce **a transformer with masked attention that enforces prior graph structure during noise prediction**.As shown in Table 1 and Table 2, **our diffusion backbone achieves smaller errors** on stationary datasets, demonstrating that the architectural design is not a trivial adaptation of DiffSTG; (2) FK steering is even further from our setting: it focuses on improving image generation quality by drawing multiple samples at inference and performing an exponentially weighted ensemble guided by a reward function. In contrast, EpiDiff performs **single-sample inference and introduces a Gaussian prior constructed from mechanistic estimates and their quantified uncertainties to steer the denoising path**. The challenges we address and the mechanism we propose are therefore fundamentally different from FK steering.

---

> > ### Author Response · Authors · 2025-11-20
> >
> > ---
> > - q1: Model performance under mis-specified mechanistic model.
> > - **R q1**: When the mechanistic model is mis-specified, the framework effectively falls back to the diffusion backbone, so performance remains close to the backbone-only model rather than being dominated by a wrong prior. In this case, the mis-specified mechanistic model fits the data poorly, leading to a flatter or more inconsistent likelihood surface; under our Laplace approximation this yields a **large parameter covariance**, which in turn induces **inflated predictive uncertainty** for the mechanistic forecast $y_M$. As a result, the uncertainty term in Eq. (12) grows and the effective guidance strength is strongly reduced, so the mechanistic prior has little influence and the diffusion backbone drives the final prediction. Our aim is not to correct an arbitrarily wrong mechanistic model, but to avoid over-trusting it: when the mechanistic fit is good, its estimate and low uncertainty provide useful structure; when it is poor or underdetermined (as in small-sample, mis-specified settings), the inflated uncertainty causes the prior to be largely ignored, which is precisely the behavior we seek.
> >
> > ---
> > [1] Wen, Haomin, et al. "Diffstg: Probabilistic spatio-temporal graph forecasting with denoising diffusion models." Proceedings of the 31st ACM international conference on advances in geographic information systems. 2023.
> >
> > [2] Singhal, Raghav, et al. "A general framework for inference-time scaling and steering of diffusion models." arXiv preprint arXiv:2501.06848 (2025).

---

> > > ### Comment · Reviewer_iLWB · 2025-11-25
> > >
> > > Thank you for your response.
> > >
> > > I still have some concerns regarding updating disease parameter for single epidemic curve. While pathogens evolve their infectivity and population develop immunity, it happens over a long period of time.
> > >
> > > In response to the other comments, I have increased my score.

---

> > > > ### Author Response · Authors · 2025-11-25
> > > >
> > > > Dear Reviewer,
> > > >
> > > > Thank you very much for your follow-up and for increasing your score.
> > > >
> > > > Regarding your concern about updating disease parameters along a single epidemic curve, our intent is **not** to model long-term changes in intrinsic pathogen properties, but to capture shorter-term changes in **effective** transmission driven by interventions, mobility, behavior, and reporting practices. In the epidemiology literature like [1,2,3], it is standard to estimate time-varying transmission rates or effective reproduction numbers from a single incidence curve at daily/weekly resolution by replacing the constant parameters $\beta,\gamma$ in Eq. (1) with time-varying $\beta(t),\gamma(t)$ fitted on sliding time windows.
> > > >
> > > > In our implementation, the parameters for each history window are **initialized from the previous window’s posterior mode** and regularized via the Laplace approximation, so they evolve smoothly over time rather than changing abruptly from window to window.
> > > >
> > > > We hope this clarifies our modeling assumption, and we would be happy to further discuss any remaining concerns.
> > > >
> > > > Best regards,
> > > >
> > > > The authors of Submission 9683
> > > >
> > > > ---
> > > > [1] Chen, Yi-Cheng, et al. "A time-dependent SIR model for COVID-19 with undetectable infected persons." Ieee transactions on network science and engineering 7.4 (2020): 3279-3294.
> > > >
> > > > [2] Cori, Anne, et al. "A new framework and software to estimate time-varying reproduction numbers during epidemics." American journal of epidemiology 178.9 (2013): 1505-1512.
> > > >
> > > > [3] Yang, Hou-Cheng, et al. "Time fused coefficient SIR model with application to COVID-19 epidemic in the United States." Journal of Applied Statistics 50.11-12 (2023): 2373-2387.

---

### Official Review · Reviewer_4uuS · 2025-11-02

**Soundness:** 3
**Presentation:** 3
**Contribution:** 4
**Rating:** 8
**Confidence:** 3

**Summary:**

This paper introduces EpiDiff, a hybrid forecasting framework designed for spatiotemporal epidemic prediction. It combines mechanistic epidemic models (like SIR) with a diffusion-model-based deep learning backbone. EpiDiff classifies and quantifies uncertainty in mechanistic estimations and uses this uncertainty to guide a spatiotemporal diffusion model during inference, adapting the influence of mechanistic models based on their confidence. Extensive evaluations on COVID-19 and influenza datasets demonstrate EpiDiff’s superiority over state-of-the-art baselines in both accuracy and robustness, especially under distribution shifts. The model also improves interpretability by quantifying and visualizing uncertainty, providing explainable forecasts that help gauge reliability. The approach is validated with comprehensive experiments, ablation studies, and sensitivity analysis, and the authors commit to releasing their code and data for reproducibility.

**Strengths:**

1. Unified framework for leveraging mechanistic epidemic models with spatiotemporal diffusion model for performance plus interpretability is novel
2. Uncertainty-aware guidance which can modulate mechanistic parameters on forecasts are very useful
3. Extensive experiments show SOTA performance over popular baselines: both traditional, deep learning and hybrid methods.
4. Case studies shoe importance of interpretability in visualizing and quantifying uncertainty
5. Ablation studies are extensive to show importance of important methodological choices

**Weaknesses:**

1. SIR dynamics are too simple for many epidemics. How scalable is it for more complex mechanistic models?
2. How sensitive are model performance w.r.t mechanistic estimates? Do these estimate correlate with other models or reports for specific epidemics?
3. How scalable is it across number of geographies? Analysis on complexity and running time would be useful

**Questions:**

See questions

---

> ### Author Response · Authors · 2025-11-20
>
> We sincerely appreciate your efforts to review our paper and provide valuable suggestions. Below we address each concern in detail.
>
> ---
> - w1: How scalable for more complex mechanistic models.
> - **R w1**:  We agree that SIR is a simplified model; but in this work our aim is to demonstrate that our **uncertainty-aware guidance mechanism can work with a mechanistic prior**, using SIR as a minimal, identifiable example. As shown in Appendix D.4 (Figure 7), **mechanistic steering itself adds almost no overhead to the diffusion backbone’s inference time**; the main additional cost of using a more complex mechanistic model lies in computing the mechanistic estimate and its uncertainty. For a mechanistic model with $n$ parameters, the per-node, per-window cost of our Laplace-based estimator scales with the parameter dimension: roughly **$\mathcal{O}(n^2)$ evaluations of the mechanistic objective for the finite-difference Hessian plus $\mathcal{O}(n^3)$ for inverting the Hessian**, on top of the cost of running the mechanistic simulator. For typical compartmental models such as SEIR or SIRD, $n$ remains small (on the order of a few parameters), so this overhead is modest and can be parallelized across regions, while the diffusion backbone continues to dominate runtime on large graphs.
>
> ---
> - w2: Model sensitivity w.r.t mechanistic estimates; correspondence with specific epidemics.
> - **R w2**: Our EpiDiff is not overly sensitive to the exact accuracy of the mechanistic estimates. As discussed in Appendix D.6 (lines 1158–1187), Figure 9 shows six representative nodes on the covid-JP dataset: in some regions (e.g., Node 1, Node 31) the SIR trajectories are relatively accurate, while in others (e.g., Node 12, Node 42) they deviate substantially and fail to capture local trends. **EpiDiff still tracks the ground truth closely**: when the mechanistic fit is poor, the posterior uncertainty increases, **automatically weakening the guidance and allowing the diffusion backbone to correct the mismatch**. Thus, the mechanistic prior acts as a **coarse but informative anchor rather than a brittle constraint**, and EpiDiff remains robust to moderate inaccuracies in mechanistic estimation.
>
> In Appendix D.7, we further provide a **practical validation** of EpiDiff by comparing the inferred nationwide $R_{\mathrm{eff}}(t)$ with the variant dynamics in Figure 10: **$R_{\mathrm{eff}}$ rises sharply from around week 25 of 2021, aligning with the rapid increase of the Delta variant in Japan, and then declines as vaccination and public health interventions take effect**, confirming that our estimates are both numerically reasonable and epidemiologically consistent.
>
>
> ---
> - w3: Scalability across number of geographies.
> - **R w3**: The method scales across geographies **approximately linearly in the number of regions $N$**. The mechanistic component is applied independently to each node and history window, so its cost grows linearly with $N$. The diffusion backbone is a graph-based Transformer whose per-layer cost scales with the number of edges; for bounded average degree, this implies $t_{\text{train}}$ and $t_{\text{inference}}$ both grow approximately linearly with $N$. We show added experiments and detailed analysis of scalability from line 1118 to line 1157.

---

> ### Author Response · Authors · 2025-11-26
>
> Dear Reviewer,
>
> Thank you again for your time and constructive comments.
>
> As we **enter the final week of the discussion period**, we wanted to follow up to ensure that our response has adequately addressed your concerns. We would greatly appreciate your feedback, as your evaluation is crucial to us.
>
> We remain available to answer any further questions you may have.
>
> Best regards,
>
> The authors of Submission 9683

---

### Author Response · Authors · 2025-11-29
**Summary of Rebuttal and Response to Reviewers**

Dear ICLR 2026 AC, SAC, and PC,

We express our sincere gratitude to all reviewers for their valuable feedback and comprehensive initial reviews. Due to the ICLR system reversion, Reviewer iLWB was the only reviewer who provided feedback to our rebuttal during the discussion period. We are particularly encouraged that they explicitly indicated our clarifications addressed their main concerns and increased their score accordingly before the incident. We appreciate the reviewers' recognition of the strengths of our work, including **good presentation and visualization** (Reviewers 4uuS, iLWB, YpmH, tsgY, xs4R), **technical soundness** (Reviewers 4uuS, iLWB, tsgY), and **contribution to hybrid models and epidemic modeling** (Reviewers 4uuS, iLWB, tsgY).

One major concern was the **scalability of the proposed EpiDiff** (Reviewers 4uuS, xs4R). We addressed this by adding a theoretical complexity analysis regarding both the mechanistic model and the number of graph nodes, as well as extra experiments on subsampled graphs to validate our analysis (from line 1118 to line 1155 in the revised paper). These results demonstrate that the runtime of our EpiDiff model scales approximately linearly with the number of nodes. We also presented dataset sizes from previous works to illustrate that we use standard epidemic datasets with epidemiological significance. Another concern was the **performance sensitivity of EpiDiff** (Reviewers 4uuS, iLWB, xs4R). We clarified that regarding the guidance scale $\tau$, we do not observe prohibitive sensitivity that would prevent practical use according to Figure 5, and it can be selected once per dataset on a validation set. We added results for EpiDiff under varying accuracy of mechanistic estimation, verifying our model's robustness to mechanistic estimations. Also, we added experiments with different forecast horizons and lookback windows, showing that the model's performance results remain stable under different experimental settings.

We have added experiments and analysis in the revised version according to the reviewers' suggestions and concerns to improve accessibility for a broader audience. Specifically:

1. Included scalability analysis of the model in Appendix D.5.
2. Included sensitivity analysis to mechanistic estimation in Appendix D.6.
3. Included a post-hoc analysis to validate that EpiDiff’s prediction results align with the emergence of the COVID Delta variant in Japan in 2021 in Appendix D.7.
4. Included sensitivity analysis to forecast horizons and lookback windows in Appendix D.8.

We respect the concern on the limited scope of robustness, uncertainty quantification, and explainability (Reviewer xs4R). However, as we emphasized in the introduction (lines 70-76), the gap we attempt to solve is **forecasting under non-stationary conditions in current epidemic modeling, as well as the explainability issues between mechanistic and data-driven components**. Reviewer xs4R acknowledged in their strengths section that we addressed these two gaps. Regarding missing data, noisy input, and causal analysis of transmission, these are not within the scope of this work, and we hope to address these issues in future works.

Finally, we are grateful that several reviewers highlighted the quality and potential impact of our work. We believe our contributions advance reliable hybrid epidemic modeling, bridging theoretical insight with practical utility. We respectfully submit that these qualities make it a valuable addition to the ICLR community and a strong candidate for acceptance.

Best regards,

The Authors of Submission 9683

---

### Meta-Review · Area_Chair_L7s2 · 2026-01-11

**Summary:**

The paper introduces a hybrid spatiotemporal epidemic forecasting framework that combines mechanistic models (SIR) with diffusion-based neural networks. One of the reviewers is positive, one is mildly positive, one is mildly negative, and two are negative. The reviewers highlight a number of concerns, most prominently regarding the significance of the technical contribution in light of previous work and the experimental evaluation. As a consequence, I am unable to recommend acceptance.

**Reviewer Concerns:**

I do not think the rebuttal have convincingly addressed the concerns regarding uncertainty quantification, explainability, and the comparison with related work.

**Reviewer Scores:**

One of the reviewers did increase their score after the rebuttal. I do not think the other reviewers would have increased their score.

---

### Decision · Program_Chairs · 2026-01-26

Reject